# Impact of Ferrous Sulfate on Thylakoidal Multiprotein Complexes, Metabolism and Defence of *Brassica juncea* L. under Arsenic Stress

**DOI:** 10.3390/plants11121559

**Published:** 2022-06-13

**Authors:** Arlene Asthana Ali, Javed Ahmad, Mohammad Affan Baig, Altaf Ahmad, Asma A. Al-Huqail, Mohammad Irfan Qureshi

**Affiliations:** 1Department of Biotechnology, Jamia Millia Islamia, New Delhi 110025, India; arlene.seven@gmail.com (A.A.A.); javedahmad12@gmail.com (J.A.); affan.mac@gmail.com (M.A.B.); 2Department of Botany, Aligarh Muslim University, Aligarh 202002, India; aahmad.bo@amu.ac.in; 3Chair of Climate Change, Environmental Development and Vegetation Cover, Department of Botany and Microbiology, College of Science, King Saud University, Riyadh 11451, Saudi Arabia; aalhuqail@ksu.edu.sa

**Keywords:** Indian mustard, arsenic, ferrous sulfate, antioxidants, thylakoidal complexes

## Abstract

Forty-day-old *Brassica juncea* (var. Pusa Jai Kisan) plants were exposed to arsenic (As, 250 µM Na_2_HAsO_4_·7H_2_O) stress. The ameliorative role of ferrous sulfate (2 mM, FeSO_4_·7H_2_O, herein FeSO_4_) was evaluated at 7 days after treatment (7 DAT) and 14 DAT. Whereas, As induced high magnitude oxidative stress, FeSO_4_ limited it. In general, As decreased the growth and photosynthetic parameters less when in the presence of FeSO_4_. Furthermore, components of the antioxidant system operated in better coordination with FeSO_4_. Contents of non-protein thiols and phytochelatins were higher with the supply of FeSO_4_. Blue-Native polyacrylamide gel electrophoresis revealed an As-induced decrease in almost every multi-protein-pigment complex (MPC), and an increase in PSII subcomplex, LHCII monomers and free proteins. FeSO_4_ supplication helped in the retention of a better stoichiometry of light-harvesting complexes and stabilized every MPC, including supra-molecular complexes, PSI/PSII core dimer/ATP Synthase, Cytochrome b6/f dimer and LHCII dimer. FeSO_4_ strengthened the plant defence, perhaps by channelizing iron (Fe) and sulfur (S) to biosynthetic and anabolic pathways. Such metabolism could improve levels of antioxidant enzymes, and the contents of glutathione, and phytochelatins. Important key support might be extended to the chloroplast through better supply of Fe-S clusters. Therefore, our results suggest the importance of both iron and sulfur to combat As-induced stress in the Indian mustard plant at biochemical and molecular levels through enhanced antioxidant potential and proteomic adjustments in the photosynthetic apparatus.

## 1. Introduction

Arsenic (As) is a naturally occurring metalloid with an inorganic form declared to be a Class I carcinogen by the International Agency for Research on Cancer (IARC), the Department of Health and Human Services (DHHS), and the Environment Protection Agency (EPA) [1]. Use of As in pesticides, herbicides and irrigation water has polluted agricultural lands. Although arsenic pollution is prevalent in South East Asia, high concentrations of As are found in the groundwater of India, Bangladesh, Chile, China, Argentina, Mexico, Hungary, Taiwan, Vietnam, Japan, New Zealand, Germany, and the United States [2]. The safe limits of As in drinking water are 0.01 mg/L according to WHO guidelines and 24 mg/kg in soil according to the US Environmental Protection Agency, but in countries like India and Bangladesh As concentrations are alarmingly high. Hence, it is not surprising to find that As has entered the food chain. For example, it is found in considerably high amounts in crops grown in As-contaminated fields in India and Bangladesh [3,4]. In such plants, As toxicity leads to compromised metabolism and, therefore, reduced plant productivity.

The role of micronutrient iron (Fe) and macronutrient sulfur (S) is pivotal in the proper functioning of crucial physiological processes. Fe acts as a cofactor in many antioxidative and redox enzymes. S is a major component of amino acids, like cysteine and methionine. Sulfur is a regulator of metal toxicity because of its involvement in the synthesis of metal chelators. Fe is also found to bind to S to form iron-sulfur [Fe-S] clusters that mediate the catalysis of crucial redox reactions during photosynthesis and respiration [5]. Iron sulfate could serve as a good source of both iron and sulfur, hence helping plants to counter stress through their ameliorative effects [6].

Considering the health-threatening consequences of As to humans and its derogative role in plant health, researchers aim to minimize the impacts of As on the health of living systems and the environment. One such way is phytoremediation, and plants belonging to the Brassicaceae family are good candidates for this technique [7]. Phytoremediation is a mammoth approach. However, it gives clues about resistant varieties of crops. Such varieties could be assisted with proper nutrition to minimize the impact of As stress. In this study, *Brassica juncea* (Indian mustard), which is a known accumulator of metals, has been investigated for its growth performance, biochemicals, antioxidants, and thylakoid membrane-bound photosynthetic apparatus. *B. juncea* L. cv. Pusa Jai Kisa is capable of tolerating comparatively high levels of As and is perhaps a better candidate among varieties of Indian mustard [8]. There are only a couple of studies on basic parameters showing that Fe-nanoparticles reduced As uptake in *Oryza sativa* [9] and decreased oxidative stress in *B. juncea* var. Pusa Jagannath [10]. Large-scale use of such nanoparticles is practically not feasible and they have exhibited a potentially negative effect on soil biota and bioaccumulation, besides causing ecotoxicity in the soil environment [11]. There is a knowledge gap on how iron sulfate affects thylakoidal multi-protein-pigment complexes (MPCs) during As stress, as reported for cadmium [5].

It is being postulated that iron-sulfate would offer better chances for the synthesis and protection of Fe-S clusters, which are essential for growth and developmental processes [12]. Therefore, the role of iron sulfate was evaluated in arsenic stress mitigation. This is the first study to reveal a good degree of unique combination of iron-sulfate-assisted amelioration, and shows the mechanism of As stress tolerance, and strengthening of the photosynthetic MPCs in *B. juncea*.

## 2. Results

### 2.1. Fresh and Dry Weights

Iron-sulfate (FeSO_4_) and arsenic (As) affected the growth of *B. juncea*, both fresh weights and dry weights of the plants. There was a statistically significant interaction between treatment and days after treatment (DAT) for fresh weight, F (3, 12) = 17.57, *p* = 0.0001, as well as for dry weight, F (3, 12) = 16.81, *p* = 0.0001, meaning the effect of treatment on fresh weight and dry weight is not the same for 7 DAT and 14 DAT. FeSO_4_ increased the fresh weight significantly by 30% and 28% at 7 days after treatment (DAT) and 14 DAT, as compared to controls. Arsenic stress (+As) decreased fresh weight significantly by 35% and 47% compared to controls at 7 DAT and 14 DAT, respectively. In the presence of As and FeSO_4_ (As + FeSO_4_) plant fresh weight showed a non-significant increase at 7 DAT and decrease at 14 DAT over the respective control plants. However, there was a significant increase over As treatment at both time points (Figure 1A). Application of FeSO_4_ increased plant dry weight significantly by 28% and 23% over control at 7 DAT and 14 DAT, respectively. As stress (+As) significantly decreased dry weight over controls by 34% at 7 DAT and by 46% at 14 DAT, respectively. In the presence of FeSO_4_ and As (As + FeSO_4_) plant dry weight increased by 14% at 7 DAT and decreased by 16% at 14 DAT, as compared to controls, together with a significant increase over As treatment (Figure 1B).

### 2.2. Root and Shoot Lengths

There was a non-significant interaction between treatment and days after treatment (DAT) for root length, F (3, 12) = 3.464, *p* = 0.0510, meaning the effect of treatment was the same for 7 DAT and 14 DAT, whereas for shoot length the interaction was significant, F (3, 12) = 4.677, *p* = 0.0219. During FeSO_4_ treatment, root length decreased significantly by 15% at both 7 DAT and 14 DAT compared to controls. Arsenic stress also decreased root length significantly by 38% and 39% over controls at 7 DAT and 14 DAT, respectively. In the combined (As + FeSO_4_) treatment, the increase in root length was significant over As treatment (Figure 2A). The shoot length was highest when the plants had FeSO_4_ added; shoot length increased by 20% and 32% over control plants at 7 DAT and 14 DAT, respectively. However, As stress significantly decreased shoot length by 33% at 7 DAT and 28% at 14 DAT. When both As and FeSO_4_ (As + FeSO_4_) were present, the decrease was 15% and 12% over controls at 7 DAT and 14 DAT, respectively, but a significant increase over As treatment was observed (Figure 2B).

### 2.3. Leaf Morphology and Leaf Area

Visual changes in leaf morphology gave a clear picture of the impact of arsenic and iron sulfate on the plant’s growth and development. During both the time points of the presented study, it was observed that the leaves were slightly larger and more pigmented (dark green) than the controls when plants were treated with FeSO_4_. At 7 DAT, the plants subjected to As stress (+As) showed a large reduction in size as well as turning yellowish near the edges (chlorosis), whereas supplementation with FeSO_4_ during As stress (As + FeSO_4_) resulted in retention of the size of leaf similar to the control set and reduced chlorosis. After the progression of time, at 14 DAT, As stress (+As) induced a more intense effect as was comprehended by the colors, as well as by the distorted shape of the leaf. This effect of As was minimized during FeSO_4_ supplementation (As + FeSO_4_) but the supplementation was not able to reverse the chlorosis effect (Figure 3). FeSO_4_ caused an increase in leaf area by 38% and 27%, As stress (+As) decreased it by 32% and 29%, and As stress in the presence of FeSO_4_ (As + FeSO_4_) decreased it by 13% and 15% at 7 DAT and 14 DAT, respectively (Figure 3).

### 2.4. TBARS Content as a Measure of Oxidative Stress

Oxidative stress was quantified in terms of thiobarbituric acid reactive substances (TBARS), for which a statistically significant interaction was obtained between treatment and days after treatment (DAT), F (3, 12) = 514.3, *p* < 0.0001. As stress (+As) increased TBARS content significantly by 53% and 125% over the respective controls at 7 DAT and 14 DAT. In combination (As + FeSO_4_) treatment, the rise in TBARS content was significantly lower than in As treatment, although still higher over controls (Figure 4).

### 2.5. Histochemical Detection of Hydrogen Peroxide in Leaf

The hydrogen peroxide formed as a result of arsenic and iron sulfate treatment was allowed to react with substrate 3,3′-Diaminobenzidine to produce dark brown insoluble adducts, appearing in the form of spots. As stress (+As), as well as iron sulfate treatment (+FeSO_4_), led to more hydrogen peroxide production than in the control, with maximum browning seen during As stress at both 7 DAT and 14 DAT. Leaves from the combined treatment of As and FeSO_4_ (As + FeSO_4_) had lesser spots as compared to As stress at both time points, indicating lesser oxidative stress (Figure 5).

### 2.6. Activities of Enzymatic Antioxidants

For superoxide dismutase (SOD) activity the interaction of treatment and days after treatment (DAT) was statistically significant, F (3, 12) = 20.87, *p* < 0.0001. SOD activity increased significantly by 19% and non-significantly by 11% over control during iron sulfate treatment (FeSO_4_) at 7 DAT and 14 DAT, respectively. It showed an increase of 37% and 31% during As stress (+As) compared to control at 7 DAT and 14 DAT, respectively. In the presence of FeSO_4_ during As stress (As + FeSO_4_), SOD activity was the highest, showing a significant increase over As treatment at both time points (Figure 6A).

Ascorbate peroxidase (APX) activity yielded a non-significant interaction between treatment and days after treatment (DAT), F (3, 12) = 2.419, *p* = 0.1169, which indicated that the treatment effect was consistent across both time points. APX activity was highest during the combined treatment of FeSO_4_ and As (As + FeSO_4_), corresponding to 71% at 7 DAT and 93% at 14 DAT, when compared to their respective controls. Iron sulfate treatment (+FeSO_4_) led to significantly increased activity of 28% at 14 DAT only. An increased activity over control was observed (45%, 69% at 7 DAT and 14 DAT, respectively) during As stress but it was significantly lesser compared to the combination (As + FeSO_4_) stress (Figure 6B).

The effect of treatments on glutathione reductase (GR) activity was significantly different for 7 DAT and 14 DAT, F (3, 12) = 9.908, *p* = 0.0014. Iron sulfate treatment (FeSO_4_) significantly increased GR enzyme activity by 55% and 52% over control at 7 DAT and 14 DAT, respectively. The activity noted during As stress (+As) accounted for an upsurge of 139% and 145% over control at 7 DAT and 14 DAT, respectively. Supplementation of FeSO_4_ along with As stress (As + FeSO_4_) further amplified GR activity to reach a prominent increase over controls (269% at 7 DAT and 288% at 14 DAT) as well as over As treatment (Figure 6C).

The interaction effect of treatment and days after treatment (DAT) on catalase (CAT) activity was significant, F (3, 12) = 14.23, *p* = 0.0003. CAT activity increased significantly as compared to control during FeSO_4_ treatment (+FeSO_4_) (73% at 7 DAT and 47% at 14 DAT). As treatment (+As) increased the enzyme activity over control significantly by 48% at 7 DAT, while at 14 DAT the increase was non-significant. The activity magnified strikingly with a significant increase of 213% and 192% over control at 7 DAT and 14 DAT, respectively, during combined treatment of As and FeSO_4_ (As + FeSO_4_), which was also significantly higher than As treatment (Figure 6D).

For Glutathione-S-transferase (GST) activity, the interaction effect of treatment and days after treatment (DAT) was statistically significant, F (3, 12) = 38.94, *p* < 0.0001. An increase in the activity of GST was noted during FeSO_4_ treatment (+FeSO_4_) significantly at 7 DAT (33% over control) and non-significantly at 14 DAT (16% over control). However, the leaves witnessed a non-significant decrease in GST enzyme activity by 23% and 18% over control at 7 DAT and 14 DAT, respectively, during As stress (+As). With the supplementation of FeSO_4_ during As stress (As + FeSO_4_) its activity rose significantly to 68% at 7 DAT and escalated to 121% over control at 14 DAT (Figure 7A).

A statistically non-significant interaction effect between treatment and days after treatment (DAT) was observed on glutathione peroxidase (GPX) activity, F (3, 12) = 1.759, *p* = 0.2083. For all the treatments, except the combined treatment (As + FeSO_4_), the changes in GPX activities between the two-time points were non-significant. During FeSO_4_ treatment (+FeSO_4_) the activity increased significantly by 70% and 64% over control at 7 DAT and 14 DAT, respectively. The addition of As (+As) elevated the activity significantly by 213% and 214% over control at 7 DAT and 14 DAT, respectively. The combined treatment of As and FeSO_4_ (As + FeSO_4_) prominently increased the GPX activity to 308% at 7 DAT and 316% at 14 DAT over controls and increased significantly over As treatment as well (Figure 7B).

The interaction effect of treatment and days after treatment (DAT) on ATP-sulfurylase (ATPS) activity was significant, F (3, 12) = 22.55, *p* < 0.0001. When iron sulfate treatment (+FeSO_4_) was provided to the plants, ATPS activity was enhanced significantly by 44% at 7 DAT and by 30% at 14 DAT over control. As stress (+As) led to a 70% and 59% significant increase in activity compared to control at 7 DAT and 14 DAT, respectively. Iron sulfate supplementation to As stressed plants (As + FeSO_4_) significantly spiked ATPS activity over control (262% at 7 DAT and 289% at 14 DAT) and over As treatment (Figure 7C).

### 2.7. Content of Non-Enzymatic Antioxidants

#### 2.7.1. Ascorbate Content

The interaction effect of treatment and days after treatment (DAT) was non-significant on ascorbate (ASA) content [F (3, 12) = 0.9199, *p* = 0.4606] and significant on dehydroascorbate (DHA) content [F (3, 12) = 28.47, *p* < 0.0001], total ascorbate (ASA + DHA) content [F (3, 12) = 17.65, *p* = 0.0001] and ratio of ASA:DHA [F (3, 12) = 514.3, *p* < 0.0001].

During iron sulfate treatment (+FeSO_4_), contents of ascorbate (ASA), dehydroascorbate (DHA) and total ascorbate (ASA + DHA) in the leaf increased by 33% (*p* < 0.001), 45% (*p* = 0.725) and 34% (*p* < 0.001), respectively, at 7 DAT and by 37% (*p* < 0.001), 49% (*p* = 0.347) and 39% (*p* < 0.001), at 14 DAT compared to the control sets. DHA content showed steep significant increases of 108% at 7 DAT and 110% at 14 DAT over control during As stress (+As). It also showed significantly high content over control (232% at 7 DAT and 412% at 14 DAT)) and over As treatment when As stress was supplemented with FeSO_4_ (As + FeSO_4_). The increases were significant over control for ASA content (13%, 15% at 7 DAT, 14 DAT, respectively), DHA content (108%, 110% at 7 DAT, 14 DAT, respectively) and total ascorbate content (22%, 25% at 7 DAT, 14 DAT, respectively) during As stress (+As), as well as for ASA content (56%, 58% at 7 DAT, 14 DAT, respectively), DHA content (232%, 412% at 7 DAT, 14 DAT, respectively) and total ascorbate content (73%, 95% at 7 DAT, 14 DAT, respectively) during combined treatment of As and FeSO_4_ (As + FeSO_4_). The ratio of ASA:DHA during FeSO_4_ treatment (+FeSO_4_) decreased non-significantly by 7% and increased non-significantly by 2% over control at 7 DAT and 14 DAT, respectively. The ratio dropped significantly by 45% at 7 DAT and by 47% at 14 DAT under As effect (+As). When FeSO_4_ was supplemented with As (As + FeSO_4_) the decrease in ratio was significant over control (−53% at 7 DAT and −69% at 14 DAT) but the decrease was non-significant over As treatment (Table 1).

#### 2.7.2. Glutathione Content

The interaction effects of treatment and days after treatment (DAT) were significant on glutathione (GSH) content [F (3, 12) = 70.79, *p* < 0.0001], oxidized glutathione (GSSH) content [F (3, 12) = 50.72, *p* < 0.0001], total glutathione (GSH + GSSG) content [F (3, 12) = 110.4, *p* < 0.0001] and ratio of GSH:GSSG [F (3, 12) = 19.08, *p* < 0.0001], implying the effect of treatments on these parameters were different for both time points.

During iron sulfate treatment (+FeSO_4_), contents of glutathione (GSH), oxidized glutathione (GSSH) and total glutathione (GSH + GSSG) in the leaf significantly increased by 43%, 16% and 37%, respectively, at 7 DAT and by 30%, 25% and 29%, over control at 14 DAT. GSH, GSSH and total glutathione showed a greater rise in content for each of the three treatments at 7 DAT. However, with the progress of time at 14 DAT the increase was significantly less than at 7 DAT. A marked increase in GSH, GSSH and total glutathione contents, by 129%, 142% and 132%, respectively, over control, was noticed during the addition of FeSO_4_ with As (As + FeSO_4_) at 7 DAT, which was also significant over As treatment. However, at 14 DAT the increase in these contents was significantly less (57%, 62% and 58% respectively, over control) compared to 7 DAT. Also, during the combined treatment (As + FeSO_4_) at 14 DAT the contents of GSH and GSH + GSSG increased significantly over As treatment, whereas for GSSG content the increase was non-significant. The ratio of GSH:GSSG increased during FeSO_4_ treatment (+FeSO_4_) by a significant 23% and by a non-significant 4% over control at 7 DAT and 14 DAT, respectively. The ratios, which were low during As treatment (−26% and −21% at 7 DAT and 14 DAT, respectively, over control), increased significantly in As-stressed plants supplemented with FeSO_4_ (As + FeSO_4_) (−5% and −1% at 7 DAT and 14 DAT, respectively, over control) (Table 2).

#### 2.7.3. Non-Protein Thiols (NPTs) Content

A statistically significant interaction effect between treatment and days after treatment (DAT) was observed on non-protein thiols (NPTs) content, F (3, 12) = 23.08, *p* < 0.0001. The NPTs increased significantly by 25% and 31% over control at 7 DAT and 14 DAT, respectively, when plants had FeSO_4_ added. The arsenic stressed condition (+As) led to a significantly higher increase in NPTs with 65% at 7 DAT and 51% at 14 DAT over control. The combined treatment of As and FeSO_4_ (As + FeSO_4_) saw an augmented effect, producing a significant increase of 119% and 87% at 7 DAT and 14 DAT, respectively, over control (Figure 8A).

#### 2.7.4. Phytochelatins (PCs) Content

The interaction effect of treatment and days after treatment (DAT) was significant on phytochelatins (PCs) content, F (3, 12) = 24.30, *p* < 0.0001. The PCs content initially decreased by 25% at 7 DAT and increased by 44% at 14 DAT over control when subjected to FeSO_4_ (+FeSO_4_), the changes being non-significant. A significant build-up of PCs was observed during As stress (+As) with increases of 76% and 91% over control at 7 DAT and 14 DAT, respectively. At 7 DAT of combination treatment (As + FeSO_4_) the PCs content increased significantly over control and decreased non-significantly over As treatment. With the progression of time (14 DAT) during the combination treatment of As and FeSO_4_ the maximum content of PCs was detected with a significant increase of 246% over control, as well as there being a significant increase over As treatment (Figure 8B).

### 2.8. Photosynthetic Pigments (Chlorophyll and Carotenoid) Content

The interaction effects of treatment and days after treatment (DAT) were significant on chlorophyll a (Chl a) [F (3, 12) = 640.8, *p* < 0.0001], chlorophyll b (Chl b) [F (3, 12) = 399.6, *p* < 0.0001] and total chlorophyll (a + b) content [F (3, 12) = 625.9, *p* < 0.0001].

The overall results of Chlorophyll a (Chl a), Chlorophyll b (Chl b) and total chlorophyll (a + b) content estimation in the leaves of *B. juncea* subjected to treatments involving arsenic and iron sulfate showed a content increase in FeSO_4_ treated set and a major decline during As stress, the effects of which were negated to a considerable extent by the addition of FeSO_4_ to As-stressed plants. When supplied with FeSO_4_ alone (+FeSO_4_) the plants’ Chl a, Chl b and total chlorophyll increased significantly by 16%, 9% and 14%, respectively, over control at 7 DAT and by 15%, 5% and 13%, respectively, over control at 14 DAT (Table 3). Leaves of plants subjected to As stress (+As) faced a significant decline of 7%, 9% and 7% over control in the content of Chl a, Chl b and total Chl (a + b), respectively, during the shorter treatment time, 7 DAT. Facing As stress for a longer duration of 14 DAT led to an intense effect on pigments and caused a significant reduction of Chl a, Chl b and total Chl (a + b) by 36%, 28% and 34% respectively, over control. Leaves of the plants supplemented with FeSO_4_ during As stress (As + FeSO_4_) showed a significant increase in Chl a, Ch b and total Chl (a + b) levels by 6%, 15% and 8%, respectively, over control at 7 DAT, which were also significant over As treatment. However, at 14 DAT, a significant decline in levels of Chl a and Chl (a + b) by 11% and 9% respectively, over control were noted, but compared to As treatment, the decline for the combined treatment (As + FeSO_4_) was significantly less, thereby establishing the positive role of iron sulfate in handling As stress.

A statistically significant interaction effect between treatment and days after treatment (DAT) was observed on carotenoid content, F (3, 12) = 474.1, *p* < 0.0001. Carotenoid content in leaves of FeSO_4_ treated plants (+FeSO_4_) significantly increased by 3% and 7% over control at 7 DAT and 14 DAT, respectively. During As stress (+As) the plants lost major contents of carotenoids i.e., they faced a significant decline over control by 22% at 7 DAT and 41% at 14 DAT. The effect of As stress in diminishing the carotenoid content was counteracted by supplementation of FeSO_4_ during As stress (As + FeSO_4_), as evidenced by a decline in carotenoid content of 5% and 3% over control, at 7 DAT and 14 DAT, respectively, which was a significantly lesser decline compared to that experienced under As treatment (Table 3).

### 2.9. Proteomic Changes in Thylakoidal Multi-Protein Complexes

#### 2.9.1. BN-PAGE of Thylakoidal Multi-Protein-Pigment Complexes

The first-dimensional Blue-Native PAGE (BN-PAGE) run of the thylakoidal membrane protein complexes resulted in complexes in their native forms being resolved based on their molecular weights in the 5–13% gradient acrylamide gel, the larger supercomplexes on the top, while the smaller complexes were on the bottom. A total of eleven such multi-protein-pigment complexes in the form of bands were observed for each of the four treatments. Variation in their band volumes could be visualized, which depicted the modulation of the thylakoid-bound photosynthetic apparatus in response to arsenic and iron sulfate. The eleven protein complexes were identified through Matrix-Assisted Laser Desorption/Ionization-Time Of Flight Mass Spectrometry (MALDI-TOF MS/MS) and subsequent database search. The complexes identified were **1**-Supercomplex 1, **2**-Supercomplex 2, **3**-Photosystem I (PSI) (RCI + LHCI), Photosystem II (PSII) core dimer, ATP synthase, **4**-PSI core, **5**-Cytochrome b6/f dimer, **6**-ATP synthase subunit, PSII core monomer, **7**-PSII sub-complex, **8**-Light-harvesting complex II (LHCII) trimer, **9**-LHCII dimer, **10**-LHCII monomer and **11**-Free proteins. The BN-PAGE gel, consisting of bands obtained in the four treatments representing the identified membrane protein complexes, is shown in Figure 9. The details of the MALDI-TOF MS/MS-based identified bands (Band no. **1** to **11**) signifying the protein complexes are listed in Appendix A.

#### 2.9.2. Band Volume Analysis

The individual band profile of the chloroplast-derived thylakoidal multiprotein complexes of *Brassica juncea*, demonstrating varied effects under the four treatment conditions, are shown in Appendix A. The comparison of the relative amounts of membrane protein complexes was brought about by ImageLab software-mediated band analysis. Each of the 11 bands in all four treatments was quantitated and compared. The change in protein relative to control was measured for each band as: [(volume_before_ − volume_after_)/volume _before_] and expressed as the percent change (Appendix A).

Changes in all complexes of thylakoids were analyzed (Figure 10A–K). The top part of the gel between 700–1000 kDa contained supercomplexes, two of which were found at 987 kDa (supercomplex 1, Band **1**) and 819 kDa (supercomplex 2, Band **2**). The content of both supercomplex 1 and supercomplex 2 during As treatment alone (+As) decreased significantly (*p* < 0.05) (−19% and −31% respectively) compared to their respective controls. However, the addition of FeSO_4_ together with As stress (As + FeSO_4_) led to an increase (7%) and decrease (−20%) of supercomplex 1 and supercomplex 2, respectively, over control, both of which were significant increases over As treatment (+As) (Figure 10A,B). The band at 553 kDa (Band **3**) comprised the entire PSI (RCI and LHCI), PSII core dimer, and ATP synthase. Individually arsenic (+As) and iron sulfate (+FeSO_4_) treatment led to significant decreases in the band volume, with As treatment reducing the content more severely than FeSO_4_ (−57% and −25%, respectively, over control). Some decline (−38%) was still evident during supplementation of FeSO_4_ with As. However, the decline was significantly less than for As treatment alone (Figure 10C).

Strong variation was observed for the band at 382 kDa (Band **4**) which was identified as PSI core. A significant reduction of band volume (−41%), as compared to control, was seen during As treatment, whereas this decreasing effect of As was significantly mitigated with the addition of FeSO_4_ (−15%). It was also noticed that FeSO_4_ treatment alone incremented the content of PSI core to some extent (7%) (Figure 10D). Analogous effects were found for the band at 355 kDa (Band **5**) which accounted for Cytochrome b6/f dimer. The addition of FeSO_4_ alone increased the band intensity (5%) compared to the control set. A significant dwindling of band intensity was observed for the As-stressed set (−44%). Indeed, supplementation of FeSO_4_ to the As-stressed set restricted the decline in content of the dimer (−24%), which was a significant increase over the As-stressed set (Figure 10E).

The band at 312 kDa (Band **6**) represented the ATP synthase subunit and PSII core monomer. Strong variation in these complexes, concerning their band volumes, was observed due to the different treatments. Arsenic (+As) and iron sulfate (+FeSO_4_) treatment individually led to significant decreases in the band volume over control, with As showing a more aggravating effect than FeSO_4_ treatment (−61% and −15% respectively). Although the band volume was reduced to half (−50%), compared to control, during supplementation of FeSO_4_ with As, the reduction was significantly limited in the As-treated set (Figure 10F). Interestingly, the band at 244 kDa (Band **7**), identified as the PSII sub-complex, showed a different modulating trend. In contrast to the above-observed bands, the PSII sub-complex showed a striking increase in band intensity (70%) over control when treated with As alone. The increased band volume receded to 52% when the As-treated set was supplemented with FeSO_4_, which was a significant reduction from that of the As-treated set (Figure 10G).

The band at 168 kDa (Band **8**) was found to be an LHCII trimer. It showed varying levels of decrement over control during the treatments. There was a 27% and 69% significant decrement in band volume compared to control during individual FeSO_4_ treatment and As treatment, respectively. The combined treatment of As and FeSO_4_ augmented the decrease to a striking 81% (Figure 10H). LHCII dimer was represented by the band at 151 kDa (Band **9**), which showed changes similar to Band **3** and Band **6**, but the modulation was less pronounced comparatively. Addition of As treatment alone led to a significant decrease in the dimer content (−38%), compared to control. The addition of FeSO_4_ alone, or supplementation of FeSO_4_ along with As, was able to retain the decrease at −17% and −24% respectively over control, showing that the reduction was significantly contained, compared to As treatment (Figure 10I).

The band at 112 kDa (Band **10**) contained an LHCII monomer. Similar to Band **7**, this band also showed an interesting trend relating to variation in band intensities across the different treatments. The monomer content enhanced significantly (67%) over control during As stress, but the enhancement receded to 14% over control when supplementation of FeSO_4_ to the As-stressed set was provided. The reduction in band volume during the combined treatment, compared to the As-treated set, was significant (Figure 10J). An analogous effect was seen for the band at 80 kDa (Band **11**), which was found to contain free proteins. The addition of FeSO_4_ alone led to a moderate increase in the number of free proteins (+15%), as compared to control. When FeSO_4_ was applied to As-stressed plants, it led to a significant increase in free protein content (+65%), but a lesser degree, as compared to the marked increase during As treatment alone (+74%). The free protein content during the combined treatment was significantly lower than during As treatment (Figure 10K).

#### 2.9.3. Second Dimension Analysis of Thylakoidal Membrane Protein Complexes

Individual thylakoidal membrane complexes that were resolved in the BN-PAGE 1st dimension were further separated into their component subunits by SDS PAGE. Each of the four lanes in the BN-PAGE gel, denoting four different treatment combinations of As and FeSO_4_, was layered onto 15% acrylamide gels after solubilizing the complexes in Sodium Dodecyl Sulfate (SDS) and run at 13 °C to separate the subunits in the second dimension. The two-dimensional (2D) gels showed protein spots that showed their variable expressions in response to control, +FeSO_4_, +As and (As + FeSO_4_) treatments (Figure 11). The gels were digitized and, thereafter, analyzed with PDQuest software for differential expression analysis of spots. The scatter plots for the control versus treated gels are shown in Appendix A. The protein spots having two-fold differential expression with those of control were marked (A to Y) on all the gels and identified through MALDI-TOF MS/MS followed by MASCOT database search selecting UniProt database. The list of proteins (along with details) identified from the 2D gels are listed in Appendix A. The identification of the subunits further confirmed the identification of the membrane protein complex bands present in the BN-PAGE (1st dimension) gel.

Of the 25 differentially expressed spots, spot ID A (Photosystem I subunit F), B (Photosystem II reaction center protein L) and M (Photosystem II protein D1), representing subunits of both complexes PSI and PSII, were found to be downregulated both during +As and (As + FeSO_4_) treatments, more strongly in the +As set. These were upregulated during +FeSO_4_ treatment compared to control. Downregulation in comparison to control was observed during As treatment with FeSO_4_ (As + FeSO_4_) or without FeSO_4_ (+As) for subunits of the complex ATP synthase, represented by spot IDs D (ATP synthase epsilon chain), K (ATP synthase subunit alpha) and R (ATP synthase subunit beta). Spot ID Q (Cytochrome f), a subunit of complex Cytochrome b6/f, was significantly downregulated during +As treatment but such a degree of downregulation was absent when As-stressed set was supplemented with FeSO_4_. Spot ID P (Chlorophyll a-b binding protein 1 or LHCII type I CAB-1), which was identified as a subunit of the complex LHCII trimer, downregulated in both the As-treated sets, with or without FeSO_4_ supplementation. On the other hand, upregulation of spot intensity was observed for Spot ID U (Chlorophyll a-b binding protein 3C or LHCII type I CAB-3C), representing LHCII monomer, during +As treatment. The upregulation of spot ID F (Ferrodoxin) and spot ID V (Ferredoxin-NADP reductase, leaf isozyme 2) was noted during FeSO_4_ treatment, with or without As stress. Spot ID W {NAD(P)H-quinone oxidoreductase subunit I} and Spot ID X {NAD(P)H-quinone oxidoreductase subunit K} were upregulated during +FeSO_4_ treatment but downregulated during +As treatment. The differential expression of each of the 25 spots is graphically shown in Appendix A and their identification as the individual proteins is shown in Appendix A.

## 3. Discussion

### 3.1. The Anti-Oxidative System of Mustard Was Not Robust Enough to Efficiently Overcome Arsenic Stress

Arsenic is known to bring morphological changes in plant growth, leaf morphology and metabolism [13,14]. We also observed adverse effects of As (250 µM) on plant metabolism, morphology, biomass accumulation and growth. The symptoms showed by leaves mimicked iron deficiency, like ‘chlorosis’, reductions in size and curling. Such morphological changes are in line with studies on As stress in plants. Such changes could be attributed to As-inhibited water nutrient uptake [15], slowing down of energy metabolism [16], and lowering of metabolism efficiency by acting on enzymes involved in photosynthesis, respiration, cell division, etc. [17,18].

Our results showed that As induced a considerable amount of oxidative stress in *Brassica juncea*. Higher production of reactive oxygen species (ROS) could be found during arsenate-arsenite state transformation and electron transfer reactions in the chloroplast and mitochondria; these ROS react with membranes and biomolecules to form peroxidation products [19,20]. Among ROS, we found a higher amount of hydrogen peroxide localized in the leaf. This might be formed by SOD action on superoxide anions or during photorespiration. However, H_2_O_2_ also acts as a mediator for ROS signaling required for stimulation of metabolic adaptations through cellular and molecular signaling, including expression regulation of genes for antioxidant enzymes, defence pathways under abiotic stress, and anabolic activities [21,22].

The activity of antioxidant enzymes, viz. SOD, APX, GPX, CAT and GR, increased significantly during As stress. An increase in SOD activity could be for scavenging ROS, as observed in many studies [23,24]. Accordingly, an increase in the activity of APX and GPX activity could be aiding in maintaining the ascorbate-glutathione pathway, to reduce H_2_O_2_ formed by SOD activity. Also, enhanced GPX expression in *Arabidopsis* and rice were correlated with abiotic stress mitigation by H_2_O_2_ homeostasis [25,26].

The activity of CAT showed a meager increase during As stress at both time points of our study, especially non-significant during a long time of stress exposure. Similar CAT activity was seen in Indian mustard exposed to heavy metal stress [27,28]. This could be attributed to inhibition of enzyme synthesis [29] or damage to CAT protein by ROS. GR activity, prominent during As stress, could be a requirement to replenish reduced glutathione and ascorbate utilized during H_2_O_2_ detoxification by the ascorbate-glutathione antioxidant cycle. Coordination between APX and GR activity, as seen in this study, is reflective of the functioning of the metal tolerance mechanism in plants as reported by others [30,31]. The same could be confirmed by the modulated levels of ascorbate and glutathione contents, and ASA:DHA and GSH:GSSG ratios. Furthermore, these molecules quench singlet oxygen and other ROS species directly as well [32].

NAD(P)H serves as a reducing power to regenerate ASA and GSH [33] and helps to limit oxidative stress. Apart from fueling the antioxidant machinery of the ascorbate-glutathione cycle, GSH (γ-Glu-Cys-Gly) is a precursor for phytochelatin synthesis and also acts in direct detoxification of xenobiotics because of its heavy metal chelating properties. It also protects proteins by maintaining their thiol groups [34]. Higher GSH content supports the process of the cell cycle whereas higher GSSG favors cell death [35]. Thus, sufficient contents of ASA and GSH, as well as higher ratios of ASA:DHA, as observed in the hyperaccumulator *Pteris vittata*, and higher GSH:GSSG ratios are favored for an effective curtailment of ROS [36,37]. Due to the significantly low ratios of reduced to oxidized non-enzymatic antioxidants due to As stress in this study, it seems that under the used arsenic level *B. juncea* could not maintain the redox-active state of the cell required for ROS homeostasis and that might have accounted for the oxidative stress seen in the form of high TBARS or malondialdehyde (MDA) and H_2_O_2_ in the leaves.

GSTs are expressed during metal stress and are responsible for conjugation and degradation of cytotoxic compounds, mediated by GSH as a co-substrate, thus preventing damage of biomolecules by these compounds. Studies have mostly observed strengthened GST activity during As stress. Transgenics of alfalfa with GST expression proved useful for phytoremediation purposes of mixed metal contaminated soil [38,39]. In the present study, however, GST activity was reduced, compared to control, during 250 µM As stress at both the time points. This may have been due to the inhibitory action of As on this enzyme, or GSH limitation.

Plants are known to regulate their sulfate assimilatory pathway according to the requirement of sulfur and cysteine to tackle heavy metal stress, as observed in *B. juncea* [40,41]. The primary reaction bringing about inorganic S assimilation is catalyzed by ATPS, the activity of which increased significantly over control at both time points in our study, however slightly lower during prolonged stress duration (14 DAT). It has been earlier documented that ATPS activity, ultimately leading to the formation of cysteine-rich compounds like GSH and PCs, aids in metal tolerance and detoxification [42,43]. The declining activity at 14 DAT in our studycould have been due to the inhibitory effect of As on the enzyme, or S limitation.

The level of NPTs elevated significantly during the short As exposure time and then decreased during longer exposure, but was still higher than controls. NPTs were found to increment during As stress in rice [44], and Zn/Cd stress in the hyperaccumulator *Arabis paniculata* [45]. Similarly, PCs content increased significantly during 250 µM As stress. The decrease in NPTs at 14 DAT and the statistically non-significant change in PCs between the two-time points may be speculative of cysteine (or sulfur) limitation, acquired for their synthesis through the S-assimilatory pathway, or due to inhibition of phytochelatin synthase after long exposure to As stress.

PCs, non-protein cysteine-rich peptides, form complexes with heavy metals like As, Cd, Cu, Zn, etc. via sulfhydryl groups of cysteine residues, for vacuolar sequestration and detoxification [46,47]. The expression of genes in the PC biosynthesis pathway, including phytochelatin synthase (PCS) that utilizes GSH as substrate, was found to boost As and Cd tolerance in *B. juncea* and other plants [48]. Nevertheless, the role of nutrient availability like sulfur and iron on PC-mediated As tolerance cannot be ruled out [18].

Estimation of photosynthetic pigment content in leaves of As-stressed *B. juncea* gave a peek into the effect exhibited by As on photosynthesis and the performance of the plant as a whole during heavy metal stress. It was observed that As stress decreased the contents of chlorophyll a, chlorophyll b, total chlorophyll, and carotenoids significantly in a time-dependent manner. The decrease in their contents may be reasoned as distortions affected in chloroplast membrane structures, like swelling and rupturing of thylakoid membranes, disorganization of thylakoid membrane-bound photosynthetic apparatus leading to changes in protein-pigment interactions, oxidation of the pigments and degradation or activity inhibition of enzymes related to pigment biosynthesis [32,49]. δ-aminolevulinic acid dehydrogenase and protochlorophyllide reductase, important for the biosynthesis of chlorophyll, have been speculated as targets of As [16,50].

Carotenoids, which perform a protective antioxidative role over the photosynthetic apparatus, receded significantly in this study, which might be due to As-mediated damages of the thylakoid membrane structure [32], thus affecting photosynthesis. Similar decreases in pigments were noted in rice and oat plants subjected to As stress [51,52]. Alternatively, increases in pigment content, as seen in some plants, can be speculated as a fortification mechanism of the plant against As stress [53]. Arsenic stress is known to contribute to Fe-deficiency, which eventually leads to lowered pigment contents [54]. The observed results of photosynthetic pigment contents seen in this study emphasized the effects of As on leaf characteristics (like chlorosis) and stunted growth of the plant.

### 3.2. Iron-Sulfate Supplement Improved Antioxidant Capacity of Brassica against Arsenic Stress

Much higher activities of enzymes, including SOD, APX, GPX, GR, CAT and GST, as well as components of the ascorbate-glutathione cycle, together with simultaneous induction of genes of the sulfate assimilatory pathway and glutathione biosynthesis, are expected to acquire the favorable redox conditions for better mitigation of As stress in *B. juncea.* Plants, with and without As (250 µM) treatment, were added with FeSO_4_ (2 mM FeSO_4_) and compared with control and As-treated plants, in an attempt to ascertain the role of iron sulfate during As stress. The changes in their physiochemical attributes at 7 DAT and 14 DAT are discussed.

Plants with combined treatment of As and FeSO_4_ exhibited better growth in terms of biomass accumulation, shoot-root length and leaf size compared to plants with As treatment. Similarly, leaf chlorosis, although more evidenced than in control plants, was reduced significantly. FeSO_4_ applied to the leaves of Fe-deficient peach and sugar-beet plants were able to promote re-greening [55]. It was observed that root lengths decreased significantly in iron sulfate-treated plants (+FeSO_4_), and more strongly in As-stressed plants (+As) compared to control, which is in agreement with Praveen et al. [10], who reported a decreased root length during treatment with iron sulfate alone or with As alone. The decrease in root length was significantly less in As-stressed plants supplemented with iron sulfate (As + FeSO_4_), compared to As-stressed plants (+As), which is a growth advantage found in our study.

Reduction in TBARS or MDA content and results of H_2_O_2_ localization in leaf proved that the oxidative stress generated by As decreased significantly with the addition of FeSO_4_. Interestingly, the leaf at 7 DAT showed intense H_2_O_2_ during FeSO_4_ treatment (+FeSO_4_). Iron sulfate treatment might have contributed to the formation and functioning of Fe-S clusters that mediate redox reactions of photosynthesis and respiration. These enhanced O_2_^●−^-forming processes in chloroplast and mitochondria might have ultimately resulted in the higher formation of H_2_O_2_ over control at 7 DAT. This can be further correlated with the significantly higher SOD activity over control observed at 7 DAT that converts the O_2_^●−^ formed to H_2_O_2_. The resultant enhanced photosynthesis and related processes could be the reason for the growth promotion of plants treated with iron sulfate (+FeSO_4_) at 7 DAT.

To understand the underlying mechanism of oxidative stress alleviation by FeSO_4_, enzymatic and non-enzymatic antioxidants were surveyed. Iron sulfate supplementation (As + FeSO_4_) resulted in SOD, APX, GPX, GR and CAT activities was significantly elevated compared to As stressed plants (+As), which might be one reason for better control of ROS generated by As. Also, it is to be noted that GST activity, which diminished during As stress compared to control, showed significantly enhanced activity on FeSO_4_ supplementation. During FeSO_4_ supplementation (As + FeSO_4_), ATPS activity and NPTs escalated to significantly higher levels time-wise compared to As-treated plants (+As), whereas the role of PCs was noticed only during 14 DAT.

Higher GSH:GSSG ratios are favored for an effective curtailment of ROS [36]. Significantly higher contents of ascorbate and glutathione, as well as the higher GSH:GSSG ratio witnessed during FeSO_4_ addition in As-stressed plants (As + FeSO_4_), as compared to As-treated plants (+As), points to better stress tolerance and, as expected, were seen to be negatively correlated with the level of ROS and MDA formed.

Photosynthetic pigment content (chlorophyll a, chlorophyll b, total chlorophyll) which were found to decline during As stress as compared to control, marked up with the addition of FeSO_4_, thereby withholding the pigment decline significantly over As-stressed plants at both the time points. Furthermore, the decline in carotenoid content seen during As exposure was significantly resisted with supplementation of FeSO_4_. Hence, the substantial upregulation of enzymatic and non-enzymatic activities/contents of *B. juncea* during combined As and FeSO_4_ treatment, correlated with less generation of H_2_O_2_ and MDA, increased pigment content and better growth in the same treatment, which may lead us to comprehend that FeSO_4_ helped tackle As-generated oxidative stress by reinforcing its anti-oxidant system.

Iron sulfate was able to fortify the anti-oxidant system, as seen in the study. FeSO_4_ is a good source of iron (Fe) as well as sulfur (S), two important nutrients required for carrying out metabolic and physiological processes like photosynthesis, and respiration, along with contributing to nitrogen and sulfur assimilation. FeSO_4_ supplementation might have helped reduce Fe-deficiency symptoms, such as leaf chlorosis affected by As stress. Fe is associated with the process of formation of two important components of the chlorophyll synthesis pathway: d-aminolevulinic acid and protochlorophyllide [56], and this role of Fe was observed as increased photosynthetic pigments when FeSO_4_ was added along with As. Fe and S are parts of iron-sulfur (Fe-S) clusters in Fe-S proteins that are known to facilitate important photosynthetic and respiratory redox reactions, take part in enzyme catalysis and help in the gene regulation process [57]. Fe ions (Fe^2+^/Fe^3+^), as well as a reduced form of sulfur, play a major role in the biosynthesis of these Fe-S clusters [6]. Ferredoxin enzymes that remain conjugated with [2Fe-2S] clusters have also been known to participate in Fe-S cluster biogenesis [57].

### 3.3. Iron Sulfate Has Multiple Roles to Play against Arsenic-Induced Stress

Besides being present in Fe-S clusters, Fe takes part (as Fe^3+^ ions) as cofactors of enzymes, one of which is the antioxidant enzyme SOD e.g., Fe-SOD, which imparts an important role in carrying out dismutation of superoxide radicals in the chloroplast [32]. In a microarray-based study, Fe-SOD was downregulated, whereas Zn/Cu SOD was upregulated, during As stress [58]. Fe is also a part of the heme group of proteins and enzymes. Ascorbate peroxidase and catalase are heme-based proteins [32]. Both these enzymes eliminate H_2_O_2_ produced by SOD activity in chloroplasts or produced via photorespiration in peroxisomes. Therefore, enhanced oxidative stress management, shown by plants supplemented with FeSO_4_ during As stress, is reflective of the role of Fe as a cofactor in augmenting the activities of these antioxidant enzymes.

FeSO_4_ is also a source of sulfate ions that plants can absorb via roots. Sulfate is fixed as sulfur by the plant in amino acids and proteins. The sulfur assimilatory pathway begins with the reduction of sulfate to sulfite by the enzyme ATP sulfurylase (ATPS), finally leading to cysteine and methionine generation [41]. Cysteine is indispensable for the formation of glutathione (GSH), phytochelatins (PCs) and other non-protein thiols (NPTs). These compounds are known to play crucial roles in detoxifying heavy metals, thereby highlighting the significance of sulfur in imparting metal stress tolerance to plants.

In this study, apart from the antioxidant enzymes, we observed induction of the sulfur assimilatory pathway, as evidenced by increased ATPS activity. Thus, the addition of sulfate ions, in the form of FeSO_4_, fulfilled the plant’s high demand for S during As stress, which is required for enhanced formation, or high contents of GSH, PCs and NPTs. Sulfite reductase, another enzyme of the pathway, functions to reduce sulfide to sulfite with the help of protein ferredoxin [41]. Fe-S conjugated ferredoxin at the photosystem I site is also essential in the regeneration of ascorbate in the thylakoid membrane, apart from its regeneration by the ascorbate-glutathione cycle [59]. Hence, Fe, apart from contributing to higher ascorbate contents, can also be envisaged as a helpful nutrient, like S, in modulating the sulfur assimilatory pathway [60].

An increase in the activities of GST and GPX when FeSO_4_ was supplemented could be because of an indirect role of Fe and S. FeSO_4_ fortified the As detoxification mechanism by the chelating ability of GSH, PCs and NPTs, thus preventing As-mediated damage of the sulfhydryl groups of GST and GPX, and, hence, their observed increased activities. Moreover, the limitation of GSH (if any) which these two enzymes need for working, seemed to be taken care of during FeSO_4_ supplementation.

### 3.4. Arsenic Stress Affected the Photosynthetic Apparatus as Evident from a Decline in Thylakoidal Membrane Multi-Protein Complexes

Arsenic stress caused severe damage to the photosynthetic apparatus, as evidenced by the significantly decreased levels of maximum components of the MPCs, viz. PSI, PSII core dimer, ATP synthase, PSI core, cytochrome b6/f dimer, ATP synthase subunit, PSII core monomer and LHCII trimer and LHCII dimer. The components which were affected to a lesser degree were the two supercomplexes: super-complex 1 and super-complex 2. Polypeptides of these complexes obtained in the second-dimensional SDS-PAGE gel were also found to be downregulated during As stress. These included PSI subunit F, PSII reaction center protein L and PSII protein D1 (subunits of PSI and PSII), ATP synthase epsilon chain, ATP synthase subunit alpha and ATP synthase subunit beta (subunits of ATP synthase), Cytochrome f (a subunit of complex cytochrome b6/f) and Chlorophyll a-b binding protein 1 or LHCII type I CAB-1 (LHCII trimer). A few of the components, however, were found in significantly increased amounts, such as PSII sub-complex, LHCII monomer and free proteins. Likewise, Chlorophyll a-b binding protein 3C or LHCII type I CAB-3C, which is a polypeptide of LHCII monomer was upregulated in the second dimensional SDS PAGE during As stress, as compared to control.

The possibility of heavy metals affecting thylakoid proteins has been explained in Cd stress-related studies. Authors have documented that metal-induced Fe deficiency is responsible for organizational changes in PSI and PSII [5,61]. The damaging effects on the membrane protein complexes during As stress might be caused by drops in the levels of the potent antioxidant, xanthophyll, brought about by Fe-deficiency [62]. As arsenic stress mimics iron deficiency like chlorosis in plants by lowering light-harvesting pigment molecules [54], As-induced Fe deficiency might be another reason for the changes observed in this study.

Arsenic stress can be envisaged as negatively influencing the aggregating property of photosystems and light-harvesting complexes to form supercomplexes due to nutritional deficiency, and, hence, the observed depletion of supercomplexes [63]. Nutrient deficiency studies in beetroot revealed that Fe-deficiency resulted in decreased light-harvesting antennae, core complexes of PSI and PSII and cytochrome b6/f [64,65], a similar result seen in this study in *B. juncea* during As stress. In Chlamydomonas, it led to the withdrawal of LHCI from PSI and the re-adjusting of antennae complexes [66]. Arsenic stress-mediated reduction of LHCs could be due to the accompanying Fe-deficiency symptoms that lead to loss of stabilizing pigments or due to repressed lhc gene expression [67,68]. A reshuffling of the antennae was seen in the present study by the decreased level of LHCII trimers and increased LHCII monomers during As stress, most likely in an attempt to create a fresh steady state, which is equipped for lesser electron transfer, or due to a decrease in chlorophyll b content as seen in the present study, and also documented in Fe-deficient mustard and spinach plants [62].

Interestingly, As stress elevated the PSII sub-complex to significantly high levels in the present study and this was in agreement with the study on *B. juncea* subjected to Fe-deficient conditions [5]. The development of the PSII sub-complex is, thus, concomitant with the degradation of the PSII core monomer.

NAD(P)H-quinone oxidoreductase subunit I and NAD(P)H-quinone oxidoreductase subunit K were downregulated during As treatment, as observed in the second dimensional SDS-PAGE. Their electron transfer activity to quinones in the chloroplastic electron transfer chain is mediated via the bound [4Fe-4S] cluster [69]. The Fe-S clusters might be provoked by the harmful ROS generated by As stress [70].

### 3.5. Iron-Sulfate Helped Retain the Thylakoidal Membrane Multi-Protein Complexes and Provided Better Stability by Readjusting Light-Harvesting Complexes during As Stress

When FeSO_4_ was added to As-stressed *B. juncea* for 14 days, the number of MPCs was higher than in plants with only As treatment, although lesser than in the controls. This substantiated the putative role of FeSO_4_ in the alleviation of As stress-mediated damage to the photosynthetic apparatus.

The components of the complexes that recovered/increased significantly by FeSO_4_ supplementation were supercomplex 1, PSI, PSII core dimer, ATP synthase, PSI core, cytochrome b6/f dimer and LHCII dimer, whereas the recovery of supercomplex 2, ATP synthase subunit and PSII core monomer were minimal, although significant. The addition of FeSO_4_ alone also led to the upregulation of supercomplex 1, supercomplex 2, PSI core and cytochrome b6/f dimer, which verified the positive role of FeSO_4_ on the photosynthetic apparatus.

The second dimensional SDS PAGE gel showed that the subunits of PSI, PSII and cytochrome b6/f viz. Photosystem I subunit F, Photosystem II reaction center protein L, Photosystem II protein D1 and Cytochrome f, which were downregulated during As stressed conditions compared to control, were found to be upregulated compared to the As-stressed plants, on the addition of FeSO_4_. It has been documented that PSI has subunits that are associated with three types of [4Fe-4S] clusters (F_X_, F_A_ and F_B_) [71]. One subunit of PSII (Photosystem II protein D1) and two subunits of the Cytochrome b6/f complex (cytochrome f and Rieske protein) are associated with Fe-S clusters as well as heme and non-heme Fe, that mediate the redox reactions of the photosynthetic electron transfer chain [72,73].

Few other components that mediate electron transfer using Fe-S clusters, such as Ferredoxin, Ferredoxin-NADP reductase (leaf isozyme 2), NAD(P)H-quinone oxidoreductase subunit I and NAD(P)H-quinone oxidoreductase subunit K, were also upregulated with FeSO_4_ in this study. Thus, it can be speculated that FeSO_4_ aided the formation of Fe-S clusters of MPCs by fulfilling the demand for micronutrient Fe which might be limiting during As stress. FeSO_4_, a source of S as well, must have improved the detoxification mechanism by forming enough GSH, PCs and NPTs as seen in this study, to protect the MPCs in the thylakoid membrane from destructive ROS formed by As.

During FeSO_4_ supplementation to As-stressed plants (As + FeSO_4_), as the decrease or breaking of PSII core monomer was significantly lesser than in the As-stressed plants, the accumulation of PSII sub-complex was also less accordingly. LHCII trimers disintegrated the most during the combined treatment but the quantity of LHCII monomers did not rise accordingly. Instead, the equilibrium shifted towards dimer formation of LHCII as the new acclimation steady-state during FeSO_4_ supplementation, which was conducive for As stress tolerance.

Thus, FeSO_4_ supplementation backed up the As stress tolerance mechanism of *B. juncea* at the molecular level by readjusting the stoichiometry of light-harvesting complexes and by maintaining the amount and structural integrity of photosynthetic membrane protein complexes.

## 4. Materials and Methods

### 4.1. Plant Growth

*Brassica juncea* (L.) Czern. (var. Pusa Jaikisan) was used as the experimental material for the present study. Authentic mustard seeds were obtained from the Indian Agricultural Research Institute (IARI), Pusa, New Delhi, India. Healthy seeds were placed in 1% Tween-20 for 5 min followed by 5 min washing in a running water stream. Seeds were then surface sterilized using 1% sodium hypochlorite/10 min and washed with sterile double distilled water (DDW) ten times. Hundreds of seeds were germinated on moist filter paper in Petri dishes, under dark conditions. Three-day-old seedlings were transferred to 250 mL beakers containing half-strength Hoagland nutrient media [74] Control and another set that received an additional 2 mM FeSO_4_ (+FeSO_4_). Plants were grown under 600 µmol photons m^−1^ s^−1^, with 16 h/8 h light/dark cycle, 22/18 °C and 75% relative air humidity in the plant growth chamber till the age of 40 days. Media was replaced every third day with fresh media. An aquarium air pump was used to aerate the nutrient solution and roots as mentioned in [5].

At the age of 40 days, mustard plants had fully expanded mature leaves that were exposed to As stress.

### 4.2. Experimental Design and Treatment

Forty-day-old *Brassica juncea* plants of both sets (Control and +FeSO_4_) were divided into two sets each making it four treatments viz. T_0_, T_1_, T_2_ and T_3_ (Appendix A), were studied at 7 (short-term) and 14 (long-term) days after treatment (DAT). Treatments were T_0_ (Control) with half-strength Hoagland nutrient media (HNM), T_1_ (HNM containing 2 mM FeSO_4_·7H_2_O, ferrous sulfate, herein FeSO_4_ or FeS or FS), T_2_ (HNM with 250 µM Na_2_HAsO_4_·7H_2_O, herein As), and T_3_ (HNM containing 250 µM Na_2_HAsO_4_·7H_2_O and 2 mM FeSO_4_, herein As + FeSO_4_). In all the experiments fully expanded third or later leaves were considered. For our study, the concentration 250 µM As (V) was arrived at according to preliminary screening experiments (data not shown). Seed germination test was performed with 0, 50, 100, 150, 200, 250, 300, 350, 400, 450 and 500 µM As (V) in which 250 µM As (V) resulted in 50% germination. The concentration of FeSO_4_ and days of treatment were arrived at according to [55] and [75], respectively.

### 4.3. Physiochemical Assessment under Arsenic Toxicity

Plants from all four treatments were studied for changes in oxidative stress, stress impacts on photosynthetic pigments, growth parameters, and antioxidant parameters.

#### 4.3.1. Estimation of the Magnitude of Oxidative Stress

To estimate changes in the content of thiobarbituric acid reactive substances (TBARS), method of Heath and Packer [76] was used. In the final step, the supernatant was read at 532 nm and its unspecific turbidity was corrected by subtracting the absorbance at 600 nm. The concentration of TBARS was calculated using the extinction coefficient (ε_532_) of 155 mM^−1^ cm^−1^.

#### 4.3.2. Histochemical Assay in Leaf

Oxidative stress in the form of hydrogen peroxide accumulated in the leaves of plants was visualized by staining the leaves with 3,3′-diaminobenzidine (DAB) following the method of Scarpeci et al. [77]. Freshly harvested leaves were dipped in a solution of DAB (1 mg mL^−1^, pH 3.8) overnight, boiled in ethanol at 80 °C for 15 min to remove the green pigments and visualized/photographed to capture reddish-brown zones.

### 4.4. Growth Parameters

#### 4.4.1. Estimation of Biomass

For estimation of fresh weight, the whole plant was harvested, washed with DDW, blot dried and then weighed on a weighing balance. For dry weight estimation, the plant was weighed after oven drying at 65 °C for 3 days or till constant weight was achieved. The weights were expressed as g per plant.

#### 4.4.2. Estimation of Shoot Length and Root Length

The shoot length of a harvested plant was measured from the tip of the shoot to the root-shoot junction with the help of a centimeter scale. The root length of the plant was measured from the root-shoot junction to the tip of the root. The lengths were expressed in centimeters (cm).

#### 4.4.3. Study of Leaf Morphology

The harvested leaves were studied and photographed to compare and visualize the effect of As on their morphology in terms of size, structure and coloration. Leaf area was calculated by the traditional method of using centimeters square on a paper sheet. Data was calculated by change compared to control. Differences were expressed as percent (%) change considering control as zero.

#### 4.4.4. In-Vitro Assay of Enzymatic Antioxidants

Enzyme kinetic assays were performed on the extracts of fresh leaves. Fresh leaf material (0.5 g) was ground in liquid nitrogen to a fine powder using a mortar and pestle. The powder was homogenized in 5 mL homogenization buffer of 100 mM potassium phosphate buffer (pH 7.5) containing 1% (*v*/*v*) Triton X-100, 1 mM ethylene diamine tetraacetate (EDTA) and 1% polyvinyl pyrrolidone (PVP). The homogenates were centrifuged at 16,099× *g* for 20 min at 4 °C. The supernatant was carefully collected with the help of a pipette and stored at −20 °C as aliquots of 500 µL till further in-vitro assessment of enzyme activities. All enzyme assays were done at 25 °C.

Superoxide dismutase (SOD) activity was assayed by the method of Dhindsa et al. [78] utilizing the ability of SOD to inhibit the photo-reduction of nitro blue tetrazolium (NBT) to the blue product formazan. One unit of SOD enzyme activity was defined as the amount of enzyme required to attain 50% inhibition of NBT reduction and expressed as Enzyme Unit (EU) mg^−1^ protein min^−1^.

Ascorbate peroxidase (APX) activity was assayed as mentioned in Qureshi et al. [79] by measuring the decrease in absorbance of ascorbate in a reaction catalyzed by APX present in the enzyme extract. One enzyme unit of APX was defined as the amount of enzyme required to convert 1 µmol ascorbate per minute at 25 °C. The APX activity was determined by using an extinction coefficient (ε_290_) of 2.8 mM^−1^ cm^−1^ and expressed as EU mg^−1^ protein min^−1^.

Glutathione reductase (GR) activity was assayed as mentioned in Connell and Mullet [80]. The enzyme activity was determined by monitoring the glutathione-dependent oxidation of nicotinamide adenine dihydrogen phosphate (NADPH). The reaction was started by adding NADPH. The decrease in absorbance at 340 nm was noted for 5 min. One enzyme unit was defined as the amount of enzyme needed to oxidize 1 µmol of NADPH per minute at 25 °C. GR activity was expressed as EU mg^−1^ protein min^−1^. The extinction coefficient for NADPH (ε_340_) used was 6.2 mM^−1^ cm^−1^.

Catalase (CAT) activity was assayed following the consumption of H_2_O_2_ by the method of Aebi et al. [81]. The reaction was started by adding 20 mM H_2_O_2_ to the mix to make a total reaction of 1 mL and it was allowed to run for 5 min at 25 °C. The decrease in absorbance of H_2_O_2_ was measured at 240 nm. One enzyme unit of CAT was defined as the amount of enzyme necessary to decompose 1 µmol of H_2_O_2_ per minute. Enzyme activity was expressed as EU mg^−1^ protein min^−1^. The extinction coefficient for H_2_O_2_ (ε_240_) used was 0.036 mM^−1^ cm^−1^.

Glutathione S-transferase (GST) activity was assayed by the method of Habig and Jakoby [82] using 1-chloro-2,4-dinitrobenzene (CDNB) as substrate. The increase in absorbance of the CDNB-conjugate formed was measured at 340 nm for 3 min at an interval of 30 s, at 25 °C. One enzyme unit of GST was defined as the amount of enzyme extract needed to form 1 nmol CDNB-conjugate per minute. Enzyme activity was expressed as EU mg^−1^ protein min^−1^. The extinction coefficient for the conjugate (ε_340_) used was 9.6 nM^−1^ cm^−1^.

Glutathione peroxidase (GPX) activity was assayed using hydrogen peroxide as a substrate according to Nagalakshmi and Prasad [83]. It was performed by monitoring the oxidation of NADPH that occurred when GR reduces GSSG that was formed by GPX activity. NADPH oxidation, recorded in a spectrophotometer for 5 min, was observed as a decrease in absorbance at 340 nm. GPX activity was expressed as EU mg^−1^ protein min^−1^. Extinction coefficient for NADPH (ε_340_) used was 6.2 mM^−1^ cm^−1^.

ATP sulfurylase activity (ATPS) activity was assayed using the protocol mentioned in [41,84]. The formation of inorganic phosphate from ATP was monitored by taking absorbance at 660 nm in a spectrophotometer. A potassium phosphate (KH_2_PO_4_) standard curve was used to determine ATPS enzyme activity, which was expressed as nmol Pi mg^−1^ protein min^−1^.

### 4.5. Estimation of Non-Enzymatic Antioxidants

#### 4.5.1. Ascorbate Content

Ascorbate (ASA), oxidized ascorbate (dehydroascorbate, DHA) and total ascorbate (ASA + DHA) content was estimated by the method of Law et al. [85].

Oxidized ascorbate (DHA) content was calculated by subtracting the amount of ascorbate (ASA) from the total ascorbate. All ascorbate content values were expressed as µmol g^−1^ DW. The ratio of ASA:DHA was also calculated.

#### 4.5.2. Glutathione Content

Glutathione (GSH), oxidized glutathione (GSSG) and total glutathione (GSH + GSSG) content was estimated by the method of Anderson [86]. The reaction was run at 25 °C for 30 min and read at 412 nm. Oxidized glutathione (GSSG) was also estimated (Total glutathione–GSH). Using GSH standard curve (0–500 µmol), values were expressed in nmol g^−1^ DW. The ratio GSH:GSSG was as well calculated.

#### 4.5.3. Non-protein Thiol (NPT) Content

Non-protein thiol content was measured according to López-Climent et al. [87]. The NPT content was expressed as nmol g^−1^ DW. The extinction coefficient for DTNB (ε_412_) used was 13,600 M^−1^ cm^−1^

#### 4.5.4. Phytochelatin (PC) Content

Phytochelatin content (PCs) in the leaves was determined by deducting the GSH content from the total NPTs. PC content was expressed as nmol g^−1^ DW.
PCs (nmol g^−1^ DW) = NPT (nmol g^−1^ DW) − total GSH (nmol g^−1^ DW)

### 4.6. Photosynthetic Pigments

Photosynthetic pigments (Chlorophyll a, Chlorophyll b and carotenoid content) were measured by the method of Hiscox and Israelstam [88] using fresh leaves (0.1 g) in 8 mL of dimethyl sulphoxide (DMSO). The volume of DMSO in the tubes was made up to 10 mL and their absorbance was measured at 480 nm, 645 nm, 510 nm and 663 nm against blank DMSO on the Beckman DU 640B spectrophotometer The chlorophyll a, chlorophyll b and total chlorophyll concentrations were determined by the formulae of Arnon [89]. Carotenoid concentration was calculated by the formula of Duxbury and Yentsch [90]. The concentrations of the photosynthetic pigments were expressed as mg g^−1^ DW.

### 4.7. Proteomic Changes in Thylakoidal Multi-Protein Complexes (MPCs)

Control (T_0_) and treated sets (T_1_: +Fe, T_2_: +As and T_3_: As + Fe) of *B. juncea* at 14 DAT were studied using Blue-Native PAGE followed by SDS PAGE (Two-dimensional BN-PAGE).

#### 4.7.1. Isolation and Processing of Thylakoid Membranes

Leaves from control and treated plants were harvested, washed with double distilled water and blot dried before proceeding for thylakoid membranes isolation using the method described in Timperio et al. [91]. Leaves were ground to a powder in liquid nitrogen and thereafter homogenized in an ice-cold buffer B1 (20 mM Tricine, pH 7.8, added with 0.3 M sucrose and 5.0 mM magnesium chloride). The homogenate was filtered through a single layer of Miracloth and centrifuged at 4000× *g* for 10 min at 4 °C. B1 buffer was added to the pellet and centrifuged again as above. The pellet was suspended in buffer B2 (20 mM Tricine, pH 7.8, added with 70 mM sucrose and 5.0 mM magnesium chloride) and centrifuged at 4500× *g* for 15 min. The final pellet obtained contained the thylakoid membranes harboring the integral membrane protein complexes.

#### 4.7.2. 2D BN-SDS PAGE

Blue-Native Polyacrylamide Gel Electrophoresis (BN-PAGE) followed by Sodium Dodecyl Sulfate Polyacrylamide Gel Electrophoresis (SDS PAGE) was carried out to study the membrane protein complexes in their native forms followed by separation of their subunits. BN-PAGE of thylakoid proteins was performed according to [5]. The isolated thylakoid membranes were added with washing buffer (330 mM sorbitol, 50 mM BISTRIS-HCl, pH 7.0, and 250 mg mL^−1^ Pefabloc as a protease inhibitor), centrifuged at 3500× *g* for 2 min at 4 °C and the pellet obtained was resuspended in 10–20 µL of 25BTH20G [25 mM BISTRIS-HCl, pH 7.0, 20% (*w*/*v*) glycerol and 250 mg mL^−1^ Pefabloc]. An equal volume of resuspension buffer containing 2% (*w*/*v*) n-dodecyl β-D-maltoside (Sigma) was added while mixing continuously.

The protein complexes were solubilized on ice for 5 min and thereafter centrifuged at 18,000× *g* for 20 min to separate the insoluble material. The supernatant was collected in a new Eppendorf tube (0.2 mL) and the concentration of proteins was measured using the Bradford method [92] using different concentrations of Bovine Serum Albumin (BSA) as standards. An equal amount of protein (100 µg) for each sample was loaded into the wells of a 5–13% (*w*/*v*) gradient acrylamide gel (0.75 mm thick) prepared using a gradient gel maker (Bio-Rad Mini PROTEAN, Hercules, CA, USA) (Appendix A). Before loading, the supernatant was mixed with 0.1 volume of Coomassie blue solution (5% Serva blue G, 100 mM Bis-Tris-HCl, pH 7.0, 30% sucrose, 500 mM e-amino-n-caproic acid) to impart a uniform negative charge to the proteins. Electrophoresis on a Protean II Bio-Rad electrophoresis system (180 × 160 mm) was performed at 4 °C for 5 h during which the voltage was gradually increased from 80 V to 200 V.

To resolve the subunits of the individual membrane complexes that were resolved in the 1st dimension, a second dimension SDS gel electrophoresis run of the proteins was performed. The lanes of each treatment of the BN-PAGE gel were excised and solubilized in a buffer containing 5% (*v*/*v*) β-mercaptoethanol and 6 M urea for 20 min at room temperature. The lanes were carefully layered on top of a 15% acrylamide gel (1 mm thick) [93] taking care as to leave no gap between the lane and the gel. The electrophoresis unit was run at a constant current of 10 mA per gel at 13 °C to separate the protein subunits till the dye front reached the bottom of the gel. The gels (1D and 2D) were visualized by staining/de-staining by the method of [94]. The gels were incubated overnight in Blue Silver stain [10% (*v*/*v*) ortho-phosphoric acid, 10% (*w*/*v*) ammonium sulfate, 0.12% Coomassie Brilliant Blue (CBB-G250) in methanol] under mild shaking; thereafter the background was de-stained by rinsing with autoclaved DDW till the bands or spots were visible.

#### 4.7.3. Image Analysis

Gels were scanned and digitized using a gel documentation system (Gel Doc^TM^, Bio-Rad, Hercules, CA, USA). Image analysis of bands was performed using ImageLab software version 6.0 (Bio-Rad, USA) which allowed band detection, quantification and normalization of band volumes. The bands (membrane protein complexes) obtained from BN-PAGE gels were subjected to comparative band volume analysis for each of the 11 bands analyzed in different treatments to obtain relative band volumes. A total of 11 bands (Band no. **1** to **11**) were sequenced to identify the membrane protein complexes. The 2D gels of each treatment were analyzed using PDQuest^TM^ software version 8.0 (Bio-Rad, USA). The protein spots were quantitated in terms of their relative volume. Spots showing more than a 2-fold change intensity were marked (Spots A to Y) and subunits of protein complexes were then identified.

#### 4.7.4. Tryptic Digestion of Proteins

In-gel tryptic digestion of proteins (contained in bands and spots) was performed according to [95]. Protein bands from 1D gel and protein spots from the 2D gels were excised and picked with a sterile scalpel and micropipette tip, respectively, in a 0.2 mL micro-tube and treated as mentioned in the source reference. The extracts from each digestion were freeze-dried in a lyophilizer and stored for further use at −80 °C.

#### 4.7.5. Peptide Mass Fingerprinting (PMF) and Protein Identification

Equal volumes of 0.1% triflouroacetic acid (TFA) and matrix (50% ACN, 0.1% TFA, and 20 g L^−1^ α-cyano-4-hydroxycinnamic acid in ultrapure water) were used to dissolve the trypsin digested peptides. Peptide mass fingerprints were generated on a MALDI-TOF/TOF MS analyzer (AB-SCIEX, TOF/TOF 5800, Applied Biosystems, USA) and the peptide mass spectra were recorded by Protein Pilot™ v.3.2 software (AB Sciex, MA, USA) through result-dependent analysis (RDA). The parameters set were as follows: MS (precursor ion) peak filtering- 800–4000 m/z interval, monoisotopic mass and mass tolerance of 50 ppm. The MS/MS (fragmentation) peak filtering was set as monoisotopic, MH^+^, minimum signal-to-noise ratio (S/N) 10 and MS/MS fragment tolerance of 0.2 Da. Protein identification was realized through a database search in the MASCOT search engine (Matrix Science, http://www.matrixscience.com, accessed on 1 May 2022) by uploading the mass spectrometry data. The parameters set during the search were as follows: database-UniProt, taxonomy-Viridiplantae (green plants), enzyme-trypsin with one missed cleavage allowed, fixed and variable modifications as Carbamidomethyl (C) and Oxidation (M) respectively, peptide charge +1, peptide mass tolerance and fragment mass tolerance as ±0.5 Da and ±0.2 Da respectively. For identification, proteins with significant probability-based scores (p < 0.05) were considered. An overview of the proteomic methodology pursued in this study is shown in Appendix A.

### 4.8. Statistical Analysis

The values reported in this work are means of three replicates with standard errors. A Repeated Measures Two-Way ANOVA in GraphPad Prism 9 and a One-Way ANOVA in IBM SPSS Statistics 28.0 was used to confirm the significance of the physiochemical assessment data and proteomics data, respectively. Comparison with the control and amongst treatments was performed using Tukey’s HSD Post Hoc tests, with *p* < 0.05 considered as the threshold for statistical significance.

## 5. Conclusions

Arsenic stress in *B. juncea* modulated the activities and levels of enzymatic and non-enzymatic antioxidants but it was not strong enough to overcome arsenic stress-mediated damages. Iron sulfate helped tackle arsenic-generated oxidative stress by reinforcing its anti-oxidant system, as well as by improving the efficiency of the sulfate assimilatory pathway that correlated with reduced levels of oxidative stress markers, increased pigment contents and better growth.

The resolving nature of BN-SDS-PAGE conveniently determined the effect of arsenic and the role of iron sulfate on thylakoidal membrane proteins of the candidate phytoremediator plant. Arsenic stress affected the photosynthetic apparatus by generating an iron deficiency condition, presumably by limiting the amounts of metabolically crucial Fe-S clusters associated with the MPCs. The MPCs were retained by iron-sulfate supplementation by retention or de novo synthesis of Fe-S clusters, heme or non-heme iron of PSI, PSII and Cytochrome b6/f as well as of Ferredoxin, Ferredoxin-NADP reductase (leaf isozyme 2), NAD(P)H-quinone oxidoreductase subunit I and NAD(P)H-quinone oxidoreductase subunit K. Iron sulfate also readjusted the stoichiometry of light-harvesting complexes as a protection mechanism. Being a source of sulfur as well, it augmented the detoxification mechanism by forming enough metal chelators to protect the thylakoid membranes from destructive ROS formed by arsenic. Thus, the use of *B. juncea* in phytoremediation of As-contaminated soils may be advanced, as found in this study, by administration of iron sulfate to the plant that strengthens its stress tolerance mechanism via enhanced antioxidant potential and proteomic adjustments in its photosynthetic apparatus.

## Figures and Tables

**Figure 1 plants-11-01559-f001:**
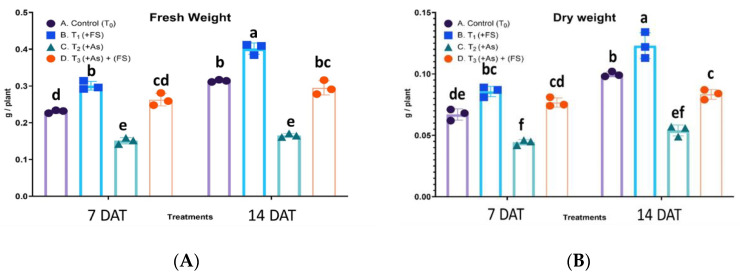
(**A**,**B**): Impact of iron sulfate (T_1_: +FeSO_4_), arsenic stress (T_2_: +As) and FeSO_4_ supplementation during As stress (T_3_: As + FeSO_4_) on fresh (**A**) and dry (**B**) weights of *Brassica juncea* studied 7 and 14 days after treatment (DAT). The values are mean ± SE and *n* = 3. Different letters on the bars indicate significant changes (*p* < 0.05) in different treatments as determined by Tukey’s HSD test. As, Arsenic; FS, FeSO_4_.

**Figure 2 plants-11-01559-f002:**
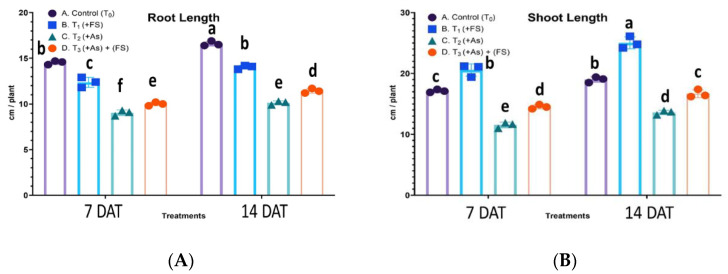
(**A**,**B**): Impact of iron sulfate (T_1_: +FeSO_4_), arsenic stress (T_2_: +As) and FeSO_4_ supplementation during As stress (T_3_: As + FeSO_4_) on the root (**A**) and shoot (**B**) lengths of *Brassica juncea* studied 7 and 14 days after treatment (DAT). The values are mean ± SE and *n* = 3. Different letters on the bars indicate significant changes (*p* < 0.05) in different treatments as determined by Tukey’s HSD test. As, Arsenic; FS, FeSO_4_.

**Figure 3 plants-11-01559-f003:**
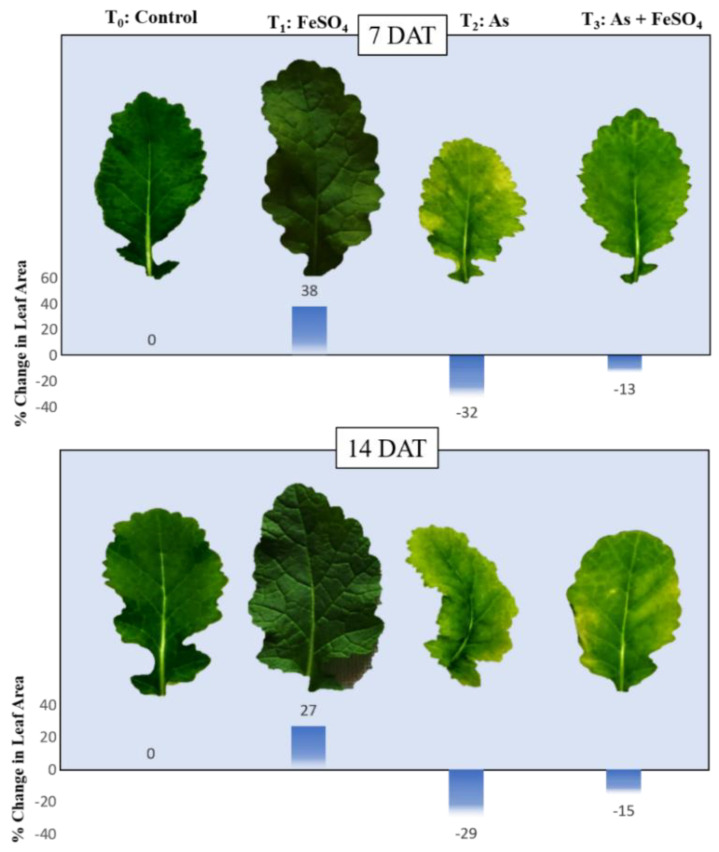
Impact of iron sulfate (T_1_: +FeSO_4_), arsenic stress (T_2_: +As) and iron sulfate supplementation during As stress (T_3_: As + FeSO_4_) on leaf morphology and percent (%) change in leaf area of *Brassica juncea* studied 7 and 14 days after treatment (DAT).

**Figure 4 plants-11-01559-f004:**
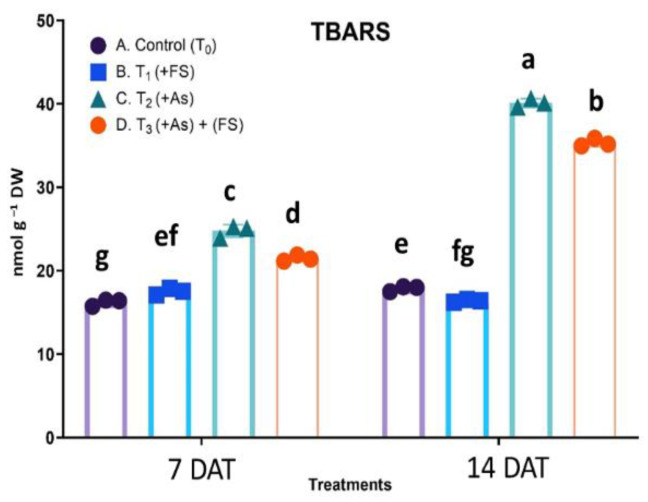
Impact of iron sulfate (T_1_: +FeSO_4_), arsenic stress (T_2_: +As) and iron sulfate supplementation during As stress (T_3_: As + FeSO_4_) on TBARS content of *Brassica juncea* studied 7 and 14 days after treatment (DAT). The values are mean ± SE and *n* = 3. Different letters on the bars indicate significant changes (*p* < 0.05) in different treatments as determined by Tukey’s HSD test. As, Arsenic; FS, FeSO_4_.

**Figure 5 plants-11-01559-f005:**
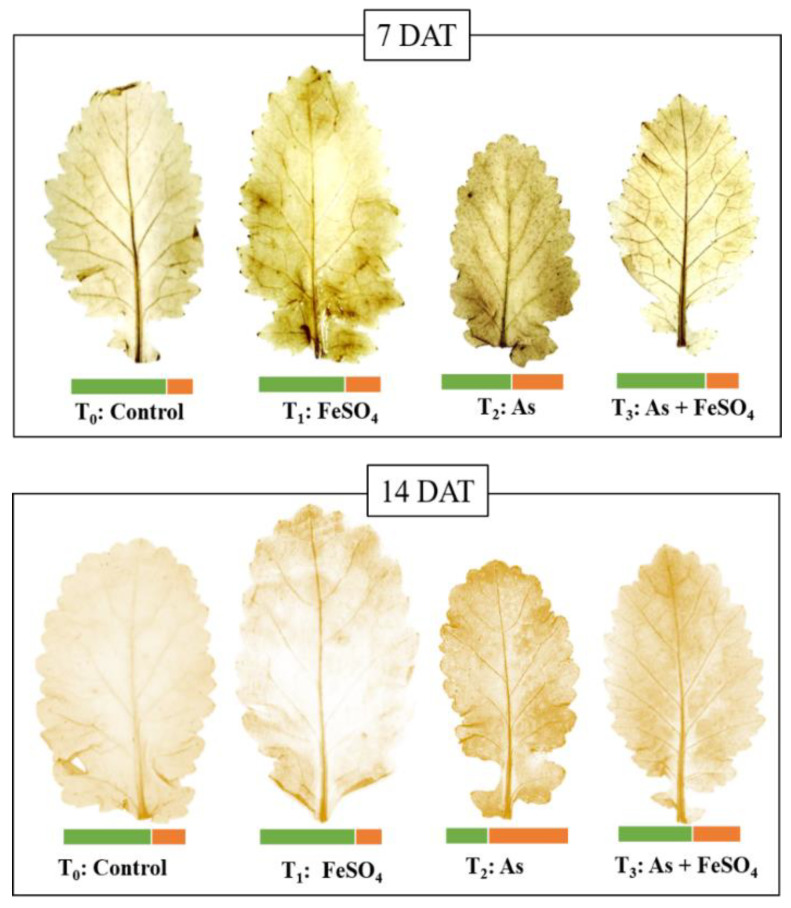
Detection of oxidative stress by hydrogen peroxide localization in the leaf of *Brassica juncea* during treatments of iron sulfate (T_1_: +FeSO_4_), arsenic stress (T_2_: +As) and iron sulfate supplementation during As stress (T_3_: As + FeSO_4_) studied 7 and 14 days after treatment (DAT). The heatmap shows a comparison of the normal scenario (green color) against H_2_O_2_ levels (orange color) in the leaf.

**Figure 6 plants-11-01559-f006:**
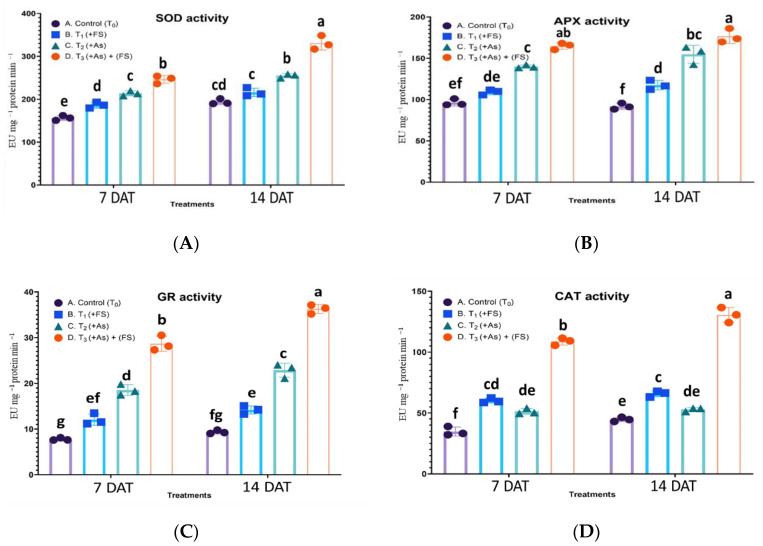
(**A**–**D**): Impact of iron sulfate (T_1_: +FeSO_4_), arsenic stress (T_2_: +As) and iron sulfate supplementation during As stress (T_3_: As + FeSO_4_) on the activity of Superoxide Dismutase (SOD, 6**A**), Ascorbate Peroxidase (APX, 6**B**), Glutathione Reductase (GR, 6**C**) and Catalase (CAT, 6**D**) in the leaf of *Brassica juncea* studied 7 and 14 days after treatment (DAT). The values are mean ± SE and *n* = 3. Different letters on the bars indicate significant changes (*p* < 0.05) in different treatments as determined by Tukey’s HSD test. As, Arsenic; FS, FeSO_4_.

**Figure 7 plants-11-01559-f007:**
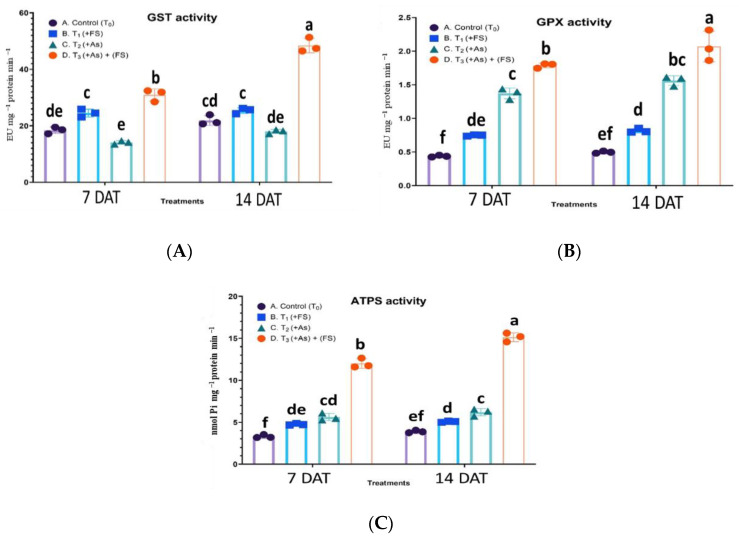
(**A**–**C**): Impact of iron sulfate (T_1_: +FeSO_4_), arsenic stress (T_2_: +As) and iron sulfate supplementation during As stress (T_3_: As + FeSO_4_) on the activity of Glutathione-S-transferase (GST, 7**A**), Glutathione Peroxidase (GPX, 7**B**) and ATP-Sulfurylase (ATPS, 7**C**) in the leaf of *Brassica juncea* studied 7 and 14 days after treatment (DAT). The values are mean ± SE and *n* = 3. Different letters on the bars indicate significant changes (*p* < 0.05) in different treatments as determined by Tukey’s HSD test. As, Arsenic; FS, FeSO_4_.

**Figure 8 plants-11-01559-f008:**
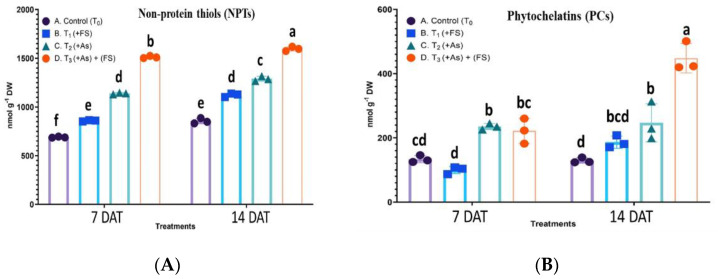
(**A**,**B**): Impact of iron sulfate (T_1_: +FeSO_4_), arsenic stress (T_2_: +As) and iron sulfate supplementation during As stress (T_3_: As + FeSO_4_) on non-protein thiols content (NPTs, 8**A**) and phytochelatins (PCs, 8**B**) content in the leaf of *Brassica juncea* studied 7 and 14 days after treatment (DAT). The values are mean ± SE and *n* = 3. Different letters on the bars indicate significant changes (*p* < 0.05) in different treatments as determined by Tukey’s HSD test. As, Arsenic; FS, FeSO_4_.

**Figure 9 plants-11-01559-f009:**
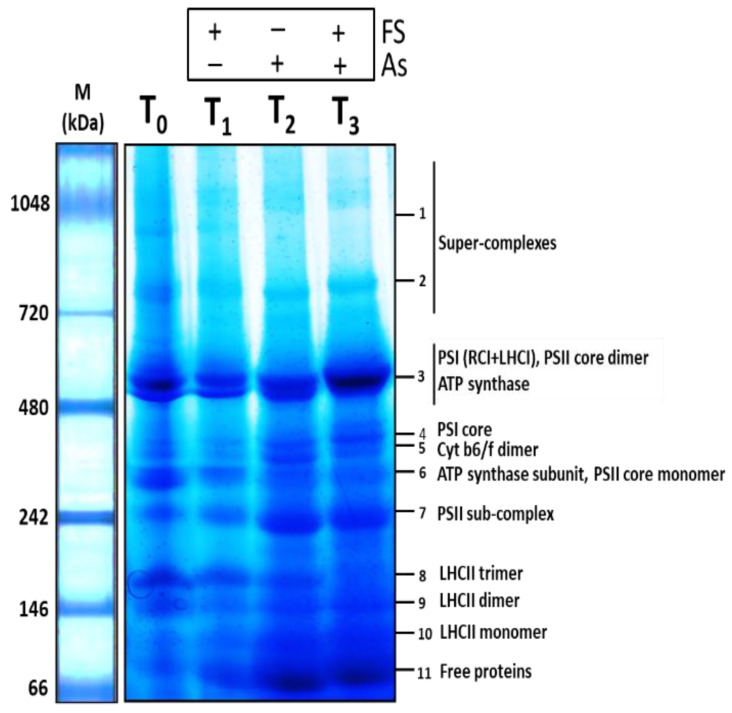
Blue-Native PAGE gel (1st dimension) showing bands representing thylakoidal membrane protein complexes in four treatment sets of *B. juncea*: T_0_ (control), T_1_ (+FeSO_4_), T_2_ (+As), T_3_ (As + FeSO_4_) at 14 DAT; M: Standard protein marker (kDa). As, Arsenic, FS, FeSO_4_.

**Figure 10 plants-11-01559-f010:**
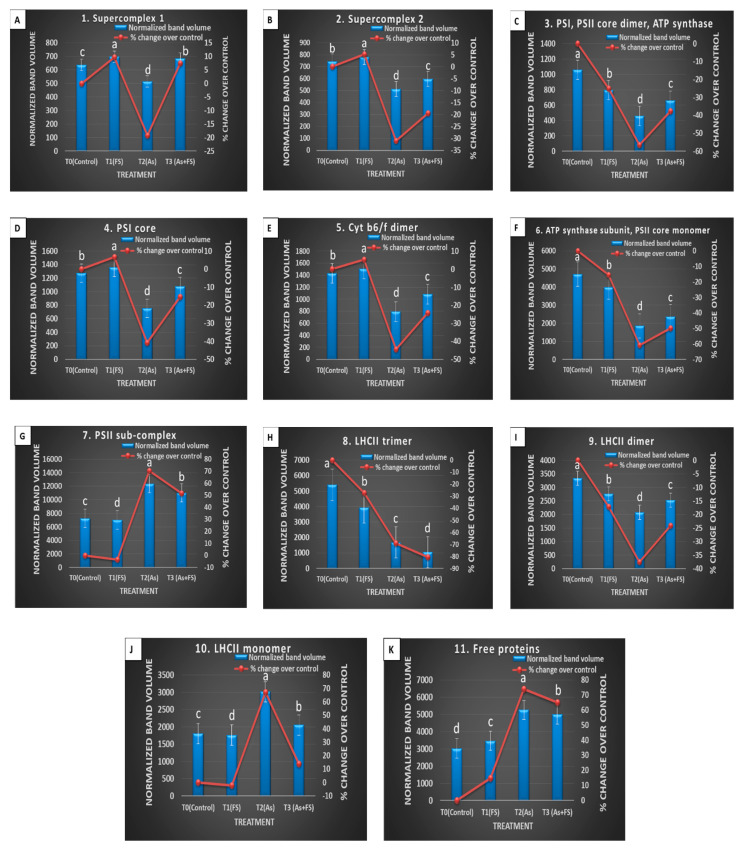
(**A**–**K**): BN-PAGE (1st dimension) gel’s band volume analysis of (**A**) Band **1**, (**B**) Band **2**, (**C**) Band **3**, (**D**) Band **4**, (**E**) Band **5**, (**F**) Band **6**, (**G)** Band **7**, (**H**) Band **8**, (**I**) Band **9**, (**J**) Band **10**, and (**K**) Band **11** mediated by ImageLab software, depicting changes in membrane protein complexes of *Brassica juncea*. The left *Y*-axis depicts normalized band volume and the right *Y*-axis depicts the % change of band volume of the treatments T1 (+FeSO_4_), T2 (+As), and T3 (As + FeSO_4_) over control at 14 DAT. The values are mean ± SE and *n* = 3. Different letters on the bars indicate significant changes (*p* < 0.05) in different treatments as determined by Tukey’s HSD test. As, Arsenic; FS, FeSO_4_.

**Figure 11 plants-11-01559-f011:**
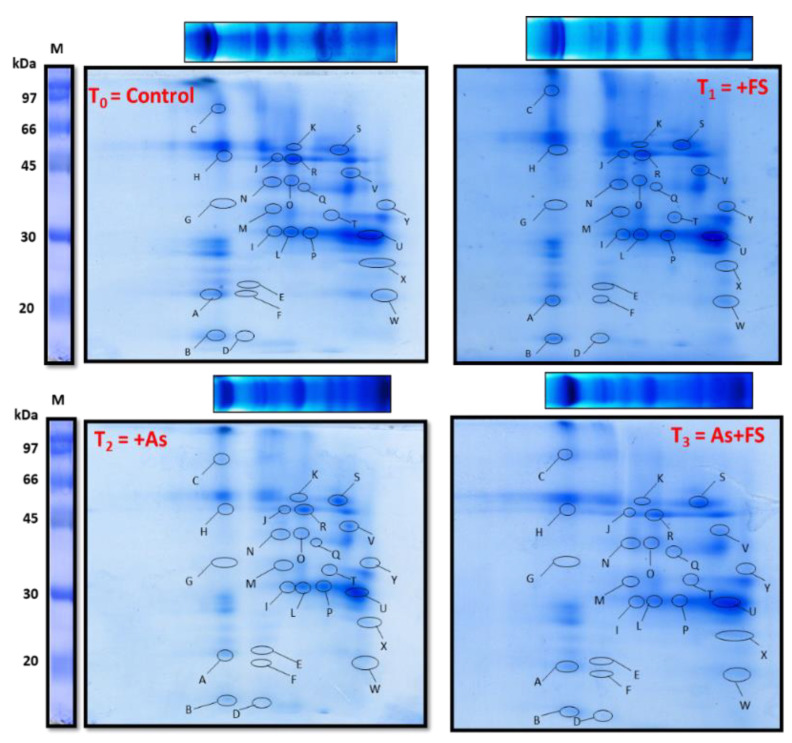
SDS-PAGE (2nd dimension) gels of thylakoid membrane proteins of *Brassica juncea* in four treatment sets T_0_ (control), T_1_ (+FeSO_4_), T_2_ (+As), T_3_ (As + FeSO_4_) at 14 DAT obtained by layering BN-PAGE gel lanes on a 15% acrylamide gel. Proteins differentially expressed and identified by MALDI-TOF mass spectrometry were marked with letters A to Y in each gel. As, Arsenic; FS, FeSO_4_.

**Table 1 plants-11-01559-t001:** Impact of iron sulfate (T_1_: +FeSO_4_), arsenic stress (T_2_: +As) and iron sulfate supplementation during As stress (T_3_: As + FeSO_4_) on contents of ascorbate (ASA), dehydroascorbate (DHA), total ascorbate (ASA + DHA) and ratio ASA:DHA in the leaf of *Brassica juncea* studied 7 and 14 days after treatment (DAT).

	Parameters	7 DAT				14 DAT			
		T_0_: Control	T_1_: +FS	T_2_: +AS	T_3_: As + FS	T_0_: Control	T_1_: +FS	T_2_: +As	T_3_: As + FS
	**ASA (µmol g ^−1^ DW**)	1.26 ± 0.01 ^e^	1.67 ± 0.03 ^c^ (+33%)	1.42 ± 0.01 ^d^ (+13%)	1.97 ± 0.02 ^b^ (+56%)	1.37 ± 0.02 ^de^	1.89 ± 0.03 ^b^ (+37%)	1.58 ± 0.02 ^c^ (+15%)	2.16 ± 0.01 ^a^ (+58%)
	**DHA (µmol g ^−1^ DW)**	0.13 ± 0.01 ^e^	0.18 ± 0.02 ^de^ (+45%)	0.26 ± 0.02 ^cd^ (+108%)	0.42 ± 0.01 ^b^ (+232%)	0.16 ± 0.01 ^de^	0.24 ± 0.02 ^cde^ (+49%)	0.34 ± 0.01 ^bc^ (+110%)	0.84 ± 0.01 ^a^ (+412%)
	**Total ascorbate (ASA + DHA) (µmol g ^−1^ DW)**	1.38 ± 0.02 ^f^	1.86 ± 0.05 ^d^ (+34%)	1.68 ± 0.03 ^e^ (+22%)	2.39 ± 0.03 ^b^ (+73%)	1.54 ± 0.03 ^ef^	2.13 ± 0.05 ^c^ (+39%)	1.92 ± 0.03 ^d^ (+25%)	3.00 ± 0.02 ^a^ (+95%)
**ASA: DHA**		9.93 ^a^	9.21 ^a^ (−7%)	5.41 ^b^ (−45%)	4.70 ^bc^ (−53%)	8.56 ^a^	8.7 5 ^a^ (+2%)	4.51 ^bc^ (−47%)	2.61 ^c^ (−69%)

The values are mean ± SE and *n* = 3. Different letters on the values indicate significant changes (*p* < 0.05) in different treatments as determined by Tukey’s HSD test. Parenthesis includes percent change over control (T_0_). As, Arsenic; FS, FeSO_4_.

**Table 2 plants-11-01559-t002:** Impact of iron sulfate (T_1_: +FeSO_4_), arsenic stress (T_2_: +As) and iron sulfate supplementation during As stress (T_3_: As + FeSO_4_) on contents of glutathione (GSH), oxidized glutathione (GSSG), total glutathione (GSH + GSSG) and the ratio of GSH:GSSG in the leaf of *Brassica juncea* studied 7 and 14 days after treatment (DAT).

Parameters	7 DAT				14 DAT			
	T_0_: Control	T_1_: +FS	T_2_: +As	T_3_: As + FS	T_0_: Control	T_1_: +FS	T_2_: +As	T_3_: As + FS
**GSH (nmol g ^−1^ DW)**	425 ± 1.3 ^f^	608 ± 1.5 ^d^ (+43%)	638 ± 1.0 ^d^ (+50%)	975 ± 0.5 ^a^ (+129%)	551 ± 2.1 ^e^	717 ± 1.5 ^c^ (+30%)	744 ± 2.0 ^c^ (+35%)	864 ± 1.6 ^b^ (+57%)
**GSSG (nmol g ^−1^ DW)**	130 ± 0.9 ^g^	151 ± 1.3 ^f^ (+16%)	265 ± 1.0 ^c^ (+104%)	315 ± 0.9 ^a^ (+142%)	175 ± 2.0 ^e^	218 ± 1.4 ^d^ (+25%)	299 ± 1.2 ^ab^ (+71%)	284 ± 2.2 ^bc^ (+62%)
**Total glutathione (GSH + GSSG) (nmol g ^−1^ DW)**	556 ± 2.2 ^f^	759 ± 2.8 ^e^ (+37%)	904 ± 2.0 ^d^ (+63%)	1290 ± 1.4 ^a^ (+132%)	726 ± 1.5 ^e^	936 ± 2.9 ^d^ (+29%)	1043 ± 3.2 ^c^ (+44%)	1148 ± 3.8 ^b^ (+58%)
**GSH:GSSG**	3.26 ^b^	4.02 ^a^ (+23%)	2.40 ^c^ (−26%)	3.09 ^b^ (−5%)	3.15 ^b^	3.28^b^ (+4%)	2.48 ^c^ (−21%)	3.11 ^b^ (−1%)

The values are mean ± SE and *n* = 3. Different letters on the values indicate significant changes (*p* < 0.05) in different treatments as determined by Tukey’s HSD test. Parenthesis includes percent change over control (T_0_). As, Arsenic; FS, FeSO_4_.

**Table 3 plants-11-01559-t003:** Impact of iron sulfate (T_1_: +FeSO_4_), arsenic stress (T_2_: +As) and iron sulfate supplementation during As stress (T_3_: As + FeSO_4_) on contents of chlorophyll a (Chl a), chlorophyll b (Chl b), total chlorophyll (a + b) and carotenoid in the leaf of *Brassica juncea* studied 7 and 14 days after treatment (DAT).

Parameters	7 DAT				14 DAT			
	T_0_: Control	T_1_: +FS	T_2_: +As	T_3_: As + FS	T_0_: Control	T_1_: +FS	T_2_: +As	T_3_: As + FS
**Chl a (mg g ^−1^ DW)**	1.435 ± 0.021 ^d^	1.661 ± 0.030 ^b^ (+16%)	1.332 ± 0.022 ^e^ (−7%)	1.517 ± 0.034 ^c^ (+6%)	1.688 ± 0.032 ^b^	1.937 ± 0.041 ^a^ (+15%)	1.086 ± 0.037 ^f^ (−36%)	1.499 ± 0.030 ^c^ (−11%)
**Chl b (mg g ^−1^ DW)**	0.394 ± 0.026 ^e^	0.429 ± 0.035 ^d^ (+9%)	0.360 ± 0.031 ^f^ (−9%)	0.452 ± 0.040 ^c^ (+15%)	0.463 ± 0.033 ^b^	0.486 ± 0.017 ^a^ (+5%)	0.334 ± 0.023 ^g^ (−28%)	0.457 ± 0.030 ^bc^ (−1%)
**Total chlorophyll (Chl a + Chl b)**	1.829 ± 0.047 ^e^	2.090 ± 0.065 ^c^ (+14%)	1.692 ± 0.053 ^f^ (−7%)	1.969 ± 0.074 ^d^ (+8%)	2.151 ± 0.065 ^b^	2.423 ± 0.058 ^a^ (+13%)	1.420 ± 0.060 ^g^ (−34%)	1.956 ± 0.060 ^d^ (−9%)
**Carotenoids (mg g ^−1^ DW)**	0.526 ± 0.021 ^c^	0.544 ± 0.028 ^b^ (+3%)	0.412 ± 0.015 ^e^ (−22%)	0.499 ± 0.021 ^d^ (−5%)	0.523 ± 0.024 ^c^	0.558 ± 0.026 ^a^ (+7%)	0.310 ± 0.001 ^f^ (−41%)	0.509 ± 0.012 ^d^ (−3%)

The values are mean ± SE and *n* = 3. Different letters on the values indicate significant changes (*p* < 0.05) in different treatments as determined by Tukey’s HSD test. Parenthesis includes percent change over control (T_0_). As, Arsenic; FS, FeSO_4_.

## Data Availability

Not applicable.

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
