# Peer review of "Impact of Ferrous Sulfate on Thylakoidal Multiprotein Complexes, Metabolism and Defence of Brassica juncea L. under Arsenic Stress"

_plants, 2022, doi:10.3390/plants11121559_

Round 1
Reviewer 1 Report
This is a very interesting project, however, the order of the chapters is not appropriate. M&M chapter follows the introduction. I recommend writing down the objectives clearly.
It looks like the experimental design was a complete randomized design but the number of reps and experimental units is unknown.
The statistical analysis is missing in the M&M chapter.
Conclusions must be based on the statistical analysis, Significant differences and treatment comparisons must be supported by usual tests.
Consider a repeated measurements model if you would like to analyze the data, and maybe a multivariate analysis as well.
Author Response
Dear Reviewer,
All authors are thankful to you for your precious suggestions and critical comments which have improved the quality of our manuscript. We have revised the MS in light of your comments and incorporated all corrections. Hope MS is now ready for acceptance
Best regards
RESPONSE:
Reviewer Report 1:
Referee: This is a very interesting project, however, the order of the chapters is not appropriate. M&M chapter follows the introduction. I recommend writing down the objectives clearly.
Response: Thank you for your appreciation. The order of the sections is as per the Journal’s format Instructions for Authors> Research Manuscript Sections. However, multiple careful readings were done to ensure the proper order. The objectives of the study have been revised to bring more clarity to the reader.
Referee: It looks like the experimental design was a complete randomized design but the number of reps and experimental units is unknown.
Response: We have added the information about replications in the Materials and Methods section.
Referee: The statistical analysis is missing in the M&M chapter.
Response: Statistical analysis has been added in the Materials and Methods section.
Referee: Conclusions must be based on the statistical analysis, Significant differences and treatment comparisons must be supported by usual tests.
Response: We have incorporated the statistical analysis. We have made sure to explain the results and discussion based on significant differences and treatment comparisons using ANOVA followed by Tukey’s HSD Post Hoc tests.
Referee: Consider a repeated measurements model if you would like to analyze the data, and maybe a multivariate analysis as well.
Response: Thank you for your suggestion. To analyze the data we have performed and incorporated the Two-Way Repeated Measures ANOVA (for physiochemical data) and One-Way ANOVA (for proteomics data) followed by Tukey’s HSD Post Hoc tests.
We Hope, the manuscript is ready for publication.
Best regards

Reviewer 2 Report
The manuscript was well prepared with clear introduction, good figure and enough discussion. There are still some questions found in the manuscript:
(1)the most important question on the study is the experimental design. How did the authors choose the suitable concentrations of Fe and As in treatments? If they were selected by experiments, Please provide related experimental data. Why the authors firstly provided additional Fe before As stress? in practice, if the As stress exist, plants have to receive its hurt even from seed germination.
(2) In the process of plant growth, is it appropriate to keep plants fit when growing in beakers containing half-strength Hoagland nutrient media without providing air or oxygen? if this is possible, the authors should give evidence that the plants did not experience hypoxia stress. otherwise, the reuslt could be not believable.
(3) the title emphasized thylakoidal multiprotein complexes, while nearly all related results have to found in supplimentary data, I suggest to adjust them.
Author Response
Dear Reviewer,
All authors are thankful to you for your precious suggestions and critical comments which have improved the quality of our manuscript. We have revised the MS in light of your comments and incorporated all corrections. Hope MS is now ready for acceptance
Best regards
RESPONSE:
Reviewer Report 2:
The manuscript was well prepared with clear introduction, good figure and enough discussion. There are still some questions found in the manuscript:
Referee: (1)the most important question on the study is the experimental design. How did the authors choose the suitable concentrations of Fe and As in treatments? If they were selected by experiments, Please provide related experimental data. Why the authors firstly provided additional Fe before As stress? in practice, if the As stress exist, plants have to receive its hurt even from seed germination.
Response: Thank you. For our study, the concentration 250 µM As(V) was arrived at according to preliminary screening experiments (data not shown). Seed germination test was performed with 0, 50, 100, 150, 200, 250, 300, 350, 400, 450 and 500 µM As (V) in which 250 µM As(V) resulted in 50% germination. The concentration of FeSO4 and days of treatment were arrived at according to El-Jendoubi et al. (2014) and Srivastava et al. (2010), respectively. This information has been updated under ‘Experimental design and treatment’.
Referee: (2) In the process of plant growth, is it appropriate to keep plants fit when growing in beakers containing half-strength Hoagland nutrient media without providing air or oxygen? if this is possible, the authors should give evidence that the plants did not experience hypoxia stress. otherwise, the reuslt could be not believable.
Response: Thank you. For our study plants were grown under 600 µmol photons m-1 s-1, with 16 h/8 h light/dark cycle, 22/18 °C and 55% relative air humidity in the plant growth chamber.
Referee: (3) the title emphasized thylakoidal multiprotein complexes, while nearly all related results have to found in supplimentary data, I suggest to adjust them.
Response: Thank you for your suggestion. We have shifted most results of thylakoidal multiprotein complexes from the Supplementary File to the main text.
We Hope, the manuscript is ready for publication.
Best regards

Reviewer 3 Report
Line 69 "(Lu 2018)" wrong citation style.
Is any statistics applied to the obtained results? In my opinion, results should be statistically processed and must be shown if any statistical significance were obtained.
Author Response
Dear Reviewer,
All authors are thankful to you for your precious suggestions and critical comments which have improved the quality of our manuscript. We have revised the MS in light of your comments and incorporated all corrections. Hope MS is now ready for acceptance
Best regards
RESPONSE:
Reviewer Report 3:
Referee: Line 69 "(Lu 2018)" wrong citation style.
Response: Thank you for bringing it to our notice. The citation style has been corrected. Furthermore, the entire manuscript has been revised.
Referee: Is any statistics applied to the obtained results? In my opinion, results should be statistically processed and must be shown if any statistical significance were obtained.
Response: We have performed and incorporated Statistical Analysis in the Material and Methods section and have explained the results and discussion of the work based on significant differences and treatment comparisons using Two-Way Repeated Measures ANOVA (for physiochemical data) and One-Way ANOVA (for proteomics data) followed by Tukey’s HSD Post Hoc tests.
We Hope, the manuscript is ready for publication.
Best regards

Round 2
Reviewer 1 Report
Fig 6B, the size of the letters is smaller than in the other graphs.
Fig 10 is difficult to see the small letters.
Author Response
Reviewer 1 Report (Round 2)
Reviewer 1: Fig 6B, the size of the letters is smaller than in the other graphs.
Author Response: Thank you for bringing it to our notice. Changes in the figure have been made.
Reviewer 1: Fig 10 is difficult to see the small letters.
Author Response: Thank you for your valuable comment. The letters in Figure 10 have been enlarged. Also, the figure has been divided into two parts for a better view and presentation with more clarity.
Best regards
Reviewer 3 Report
Dear authors,
Line 80 the sentence "As stress decreased fresh weight signif- 80 icantly by 35% and 47% compared to controls at 7 DAT and 14 DAT, respectively" is not clear.
Author Response
Reviewer 3 Report (Round 2)
Dear authors,
Line 80 the sentence "As stress decreased fresh weight signif- 80 icantly by 35% and 47% compared to controls at 7 DAT and 14 DAT, respectively" is not clear.
Author Response: Thank you for your valuable comment. The sentence has been corrected.
Best regards
